# Seeing Sound, Hearing Sight: Uncovering Modality Bias and Conflict of AI models in Sound Localization

**Yanhao Jia[1], Ji Xie[1], S Jivaganesh[1], Hao Li[2], Xu Wu[1,3], Mengmi Zhang[1†]**

[1] Nanyang Technological University, Singapore
[2] Peking University, China    [3] Shenzhen University, China
† Corresponding author; address correspondence to `mengmi.zhang@ntu.edu.sg`

## Abstract

Imagine hearing a dog bark and instinctively turning toward the sound—only to find a parked car, while a silent dog sits nearby. Such moments of sensory conflict challenge perception, yet humans flexibly resolve these discrepancies, prioritizing auditory cues over misleading visuals to accurately localize sounds. Despite the rapid advancement of multimodal AI models that integrate vision and sound, little is known about how these systems handle cross-modal conflicts or whether they favor one modality over another. Here, we systematically and quantitatively examine modality bias and conflict resolution in AI models for Sound Source Localization (SSL). We evaluate a wide range of state-of-the-art multimodal models and compare them against human performance in psychophysics experiments spanning six audiovisual conditions, including congruent, conflicting, and absent visual and audio cues. Our results reveal that humans consistently outperform AI in SSL and exhibit greater robustness to conflicting or absent visual information by effectively prioritizing auditory signals. In contrast, AI shows a pronounced bias toward vision, often failing to suppress irrelevant or conflicting visual input, leading to chance-level performance. To bridge this gap, we present EchoPin, a neuroscience-inspired multimodal model for SSL that emulates human auditory perception. The model is trained on our carefully curated AudioCOCO dataset, in which stereo audio signals are first rendered using a physics-based 3D simulator, then filtered with Head-Related Transfer Functions (HRTFs) to capture pinnae, head, and torso effects, and finally transformed into cochleagram representations that mimic cochlear processing. To eliminate existing biases in standard benchmark datasets, we carefully controlled the vocal object sizes, semantics, and spatial locations in the corresponding images of AudioCOCO. EchoPin outperforms existing models trained on standard audio-visual datasets. Remarkably, consistent with neuroscience findings, it exhibits a human-like localization bias, favoring horizontal (left–right) precision over vertical (up–down) precision. This asymmetry likely arises from HRTF-shaped and cochlear-modulated stereo audio and the lateral placement of human ears, highlighting how sensory input quality and physical structure jointly shape precision of multimodal representations. All code, data, and models are available here.

## 1 Introduction

Sound source localization (SSL) is the task of identifying the spatial origin of a sound within a visual scene. It plays a fundamental role in both biological perception [1] and artificial intelligence (AI) [2], enabling systems to connect what they hear with what they see. In natural environments, visual inputs interact with auditory cues, often dominating or recalibrating sound perception—a phenomenon exemplified by classic cross-modal illusions [3]. In practice, accurate SSL is critical for

39th Conference on Neural Information Processing Systems (NeurIPS 2025).

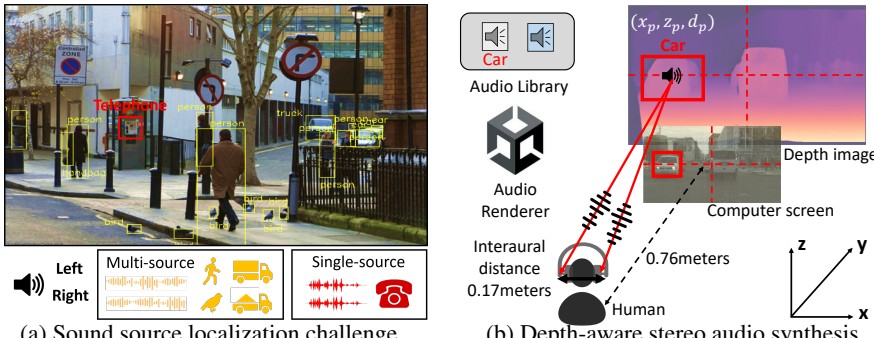

(a) Sound source localization challenge      (b) Depth-aware stereo audio synthesis

Figure 1: **(a) Sound source localization challenge in naturalistic images.** In the multi-source scenario (orange panel), multiple objects in the scene—such as pedestrians, birds, cars, and trucks (highlighted by yellow boxes)—emit sounds, whereas in the single-source scenario (red panel), only one object, a telephone (red box), produces sound. In both settings, the task is to localize all sounding objects based on two-channel stereo audio. To allow systematic and controllable benchmarking of human and AI performance, we focus on the single-source localization task (red panel), which remains challenging due to scene clutter, occlusions, and ambiguous visual cues. **(b) Depth-aware stereo audio synthesis.** In the 3D simulator, a human listener (interaural distance: 0.17m) is placed at the origin, facing the RGB image on the screen. This image and its depth image are aligned in the same direction. Using a spatial audio renderer and audio from our library, stereo audio of the target car (red box) in the RGB image can be synthesized (see **Sec 2.1**).

real-world systems such as autonomous vehicles anticipating hazards [4], assistive technologies for visually impaired users [5], and robots operating in human-centered environments [6, 7, 8, 9, 10]. These applications demand robust auditory inference under noisy, cluttered, and ambiguous sensory conditions. Consider a real-life example in **Fig. 1a** depicting a busy traffic intersection: cars honk, people converse, birds chirp, and an ambulance siren approaches from behind a building. Humans must rapidly detect and localize the siren despite competing sounds, partial visual occlusion, and environmental noise. Such everyday scenes highlight the challenges of SSL: resolving ambiguous visual or auditory cues, handling conflicting signals across modalities, and focusing attention on the most relevant source amidst distractions. Building AI systems capable of performing SSL tasks in these conditions remains an open problem.

Recent AI research has developed multimodal models for SSL [11, 12, 13, 14, 15, 2, 16]. Methods such as contrastive learning [17, 18, 19] aim to align audio-visual embeddings by maximizing the similarity between matching pairs and minimizing it between mismatched ones. Other approaches leverage cross-modal attention [20, 21, 22] in transformer architectures [23, 24] to allow sound features to dynamically query relevant visual regions. Some works [25, 26, 27] also incorporate object priors, leveraging knowledge of what typically makes sound to guide localization. However, despite these advances [28], little is known about how such models behave when the modalities conflict or when biases emerge—such as favoring visual cues over auditory ones.

To address this gap, we systematically evaluate model behavior under six controlled audio-visual conditions: (1) Congruent — audio and visual cues align in both semantics and location; (2) conflicting visual cues, where the visual scene misleads localization; (3) absent visual cues, where the sounding object is completely occluded in the visual scene; and (4) vision-only and (5) audio-only conditions, where either vision or audio is entirely omitted. Moreover, similar to the cocktail party problem [29, 30], we further extend the single-source stereo audio to (6) the multi-instance SSL [31], where multiple instances of the same semantic category are present, potentially distracting the model from correctly localizing the target instance. These manipulations allow us to probe how models resolve cross-modal ambiguity and characterize their reliance on each modality. To benchmark these model behaviors, we additionally conduct human behavioral studies under the same experimental conditions. Results show a clear performance gap: humans consistently outperform AI models in handling both congruent and incongruent conditions.

While numerous SSL datasets [32, 33, 34] exist, they often suffer from limitations that hinder robust multimodal alignment in AI models. Typically, these datasets [35, 36, 37, 38, 39, 40, 41] consist of scenes with a single, large, centrally placed sounding object, making them vulnerable to visual shortcut learning, where models perform well without truly integrating audio information [42, 43]. Others [44, 45, 46] are constructed by pairing unrelated images and sounds from independent

datasets [47, 48, 49, 50, 51, 52, 53, 54, 55], leading to weak cross-modal entanglement. More recent efforts [56, 57, 58] synthesize audio based on physics-informed rules [59, 60] or generative AI models [61, 62], but they still frequently rely on mono audio, neglecting the richer spatial cues provided by stereo audio signals.

Inspired by neuroscience findings highlighting the role of inter-channel differences in spatial hearing [63], we propose a method that leverages 3D simulation engines to generate stereo audio from images by integrating separate image and sound datasets. Unlike neuroscience approaches that require physically setting up microphones in real-world spaces, which is both time-consuming and costly, our simulation-based method offers a scalable and efficient alternative for producing spatialized audio paired with complex visual scenes. With this method, we contribute a large-scale AudioCOCO dataset, comprising 28,224 image-audio pairs with ground truth annotations. By simulating stereo audio that adheres to physical principles of sound propagation and spatial cues, AudioCOCO provides realistic and diverse audio-visual scenes.

Human sound localization relies on a cascade of acoustic transformations shaped by the ear, head, and torso. Direction-dependent spectral notches introduced by the pinna help resolve elevation and front–back ambiguities [1, 64, 65]. These cues are formalized in the Head-Related Transfer Function (HRTF), which encodes interaural time (ITD) and level differences (ILD) alongside fine-grained spectral features. At the cochlea, sounds are further decomposed into frequency-selective channels that preserve ITD/ILD while transforming HRTF-induced modulations into neural representations.

To model these biological mechanisms, we introduce EchoPin, a neuroscience-inspired model for SSL. EchoPin pre-processes stereo audio using Head-Related Transfer Function (HRTF)–based filtering and cochleagram representations derived from Equivalent Rectangular Bandwidth (ERB) filters. These designs capture the tonotopic organization and temporal dynamics of the auditory periphery [66, 67] more faithfully than conventional mel-spectrograms [68, 69, 70]. EchoPin then employs dual encoders to jointly process audio and visual inputs, trained with contrastive learning on our AudioCOCO dataset. Experimental results suggest that EchoPin outperforms existing models. Notably, without any human behavioral supervision, EchoPin reproduces a human-like horizontal localization bias [71], an emergent property that was not prominent in previous AI systems. We attribute this to EchoPin's 3D stereo audio pipeline, which integrates interaural spacing, HRTF filtering, and ERB-based cochlear processing to jointly enhance sensory fidelity and multimodal alignment. We highlight our key contributions below:

1. We introduce a unified framework to systematically benchmark audio-visual localization models under modality conflicts, absence, and misalignment. We propose a scalable pipeline that synthesizes two-channel stereo audio for static images via 3D simulation, and construct a large-scale, naturalistic, and spatially grounded audio-visual dataset, named as AudioCOCO.

2. We design and conduct psychophysics experiments to assess human strategies in resolving audio-visual conflicts, providing a strong baseline for AI-human comparison. We provide a detailed analysis of modality conflicts and biases in existing SSL models and humans, highlighting key performance and behavioral differences under challenging multimodal conditions.

3. We introduce EchoPin, a neuroscience-inspired model trained on our curated AudioCOCO dataset. It features a dual-encoder architecture that processes stereo audio–visual pairs. The stereo audio is pre-processed using HRTF-based spatial filtering and cochlear-inspired frequency decomposition. Experimental results show that precise audio–visual alignment emerges from high-fidelity sensory inputs and biologically grounded ear-structure priors. EchoPin not only achieves superior localization accuracy but also exhibits human-like localization biases, favoring horizontal over vertical precision.

## 2 Experiments

### 2.1 AudioCOCO dataset

**Image selection.** We use the MSCOCO [47] dataset for its broad coverage of everyday objects and apply our selection criteria below to all images from the standard training and test sets. From its dataset annotations, we manually select 12 audible object categories encompassing humans, animals, vehicles, and electronic devices. Recognizing that larger objects may be easier to detect and localize, we further categorize sounding objects by their relative size in the image. Object size is defined as the ratio of the object's segmentation mask area to the total image area, independent of their

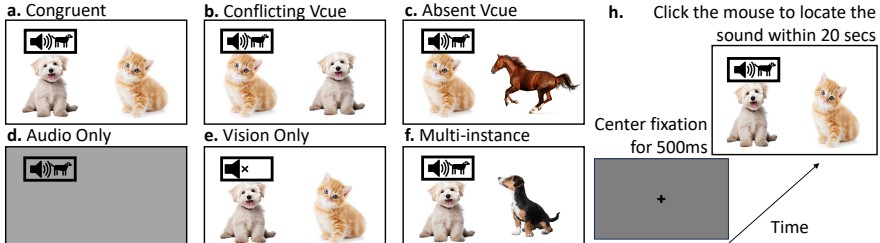

Figure 2: **Overview of congruent and manipulated vision-audio conditions in our UniAV framework and task schematic**. An example of the vision-audio congruent condition (a) is shown, where a dog sound is played (black box) with a matching visual source. Visual and auditory modifications for the other five experimental conditions (b-f) are also displayed. See **Sec 2.2** for more details. (h) Each trial began with a fixation cross (500 ms), followed by the presentation of an image-audio pair from either of the six conditions (a-f). Participants were instructed to use the computer mouse to click on the perceived location of the sound source within 20 seconds.

real-world scale. We define three size bins: Size1 (0–5%), Size2 (5–15%), and Size3 (15–30%). Objects occupying more than 30% of the image are excluded, as they make the localization task trivially easy. To maintain class balance, we limit the number of images per object size per category to 150 within each training or test set. For experimental control, each image contains only one target sounding object, without any other instances of the same category. For the multi-instance SSL (**Sec 2.2**), we select 150 images per object size per category, each containing 2–5 instances of the same semantic category, with exactly one designated as the sounding target. See **Supp Fig. S1**, **Supp Fig. S2**, and **Supp Fig. S3** for the distribution of image counts by category and their target spatial locations in both the training and test sets. After applying the selection criteria, we randomly sample 4,953 qualified images from the MSCOCO training set for training, and 5,500 images from the official test set—comprising 2,840 single-object and 2,660 multi-object scenes—for testing.

**Audio selection.** Prior work often pairs MSCOCO images with external audio datasets like VGGSound [49] or FSDnoisy18k [50], but these audio sources often lack spatial and temporal consistency. For example, moving sound sources (e.g., a dog running across the video frames) cause spatial shifts that make them unsuitable for generating realistic stereo audio on static images. Additionally, recordings may include background noise or mixed sounds from multiple objects, reducing semantic clarity and overall quality. To address these issues, we apply three filtering steps to the VGGSound videos, retaining 3,533 clips from the standard training set and 727 clips from the standard test set that contain high-quality audio, spatially and semantically aligned with the visual content. See **Supp Sec. S1**, **Supp Fig. S4**, and **Supp Fig. S5** for detailed filtering steps and results.

Next, we randomly pair the selected MSCOCO images with high-quality audios from VGGSound to construct the AudioCOCO dataset. The dataset comprises 9,360 audio–image pairs in the training set and 18,864 pairs in the test set, which are further divided into six experimental conditions (**Sec. 2.2**).

**Depth-aware stereo audio synthesis.** To generate spatialized stereo audio for their corresponding sounding objects on static images, we employ Unity [72] as our 3D simulation engine, which allows us to control precise object locations and simulate realistic stereo sound based on the spatial layout of visual scenes. As illustrated in **Fig. 1b**, we define a Cartesian coordinate system within Unity. The listener is positioned at the origin (0, 0, 0). The computer screen displays the image to listener, is placed parallel to the x–z plane and aligned along the positive y-axis, with a fixed physical distance of 0.76 meters from the listener in the human psychophysics experiment (**Sec 2.3**). To estimate the depth of objects within the 2D image, we utilize the DepthAnything model [73], which outputs a relative depth map with values ranging from 0 to 10. However, without access to the original camera intrinsics of MSCOCO images, determining absolute scene scale in Unity is nontrivial. To address this, we normalize the depth values $d_p$ of the image as $d_p^{\text{norm}} = d_{p,\max} - d_p$ where $d_{p,\max}$ are the maximum depth values in the image. Sound loudness decreases logarithmically with distance from the listener. To maintain sound amplitudes within a comfortable and perceptible range, we rescale the normalized depth by 0.5. The final y-coordinate in Unity $y_u$ for the sounding object is computed as $y_u = d_p^{\text{norm}}/2 + 0.76$. For the $x$ and $z$ coordinates of the sounding object in Unity, we map the object's pixel location from the 2D image to the physical dimensions of the monitor. Given that the display has a resolution of 90 pixels per inch, we compute $x_u = x_p/90$ and $z_u = z_p/90$, where $x_p$ and $z_p$ are the pixel coordinates of the target object's center in the image.

Once the sounding object's 3D position $(x_u, y_u, z_u)$ is established in Unity, we place a static audio source at this location. Using an interaural distance of 0.17 meters, reflecting the typical distance between human ears, Unity simulates realistic two-channel stereo sound based on the spatial relationship between the listener and the sound source. This procedure allows us to synthesize spatially grounded, depth-aware stereo audio for each image-sound pair.

## 2.2 Experimental conditions in the AudioCOCO test set

As shown in **Fig. 2**, the AudioCOCO test set includes six experimental conditions, totaling 18,864 image–audio pairs to systematically probe modality biases and conflicts in humans and AI models. Each condition contains 2,900 pairs, except MultiInstLoc, which includes 4,364. All models are trained only on congruent conditions to evaluate generalization, while these six conditions are used exclusively for testing.

**Audio-visual Congruent (Congruent)** represents the ideal scenario where both the audio semantics and localization align perfectly with the corresponding visual target's semantics and location. This should serve as the upper bound for performance in both humans and AI models. For instance, as shown in **Fig. 2(a)**, a dog sound is played at the same location as the dog in the image.

**Conflicting Visual Cue (ConflictVcue)** examines the scenario where the semantics of both the visual and audio cues belong to the correct category but are spatially misaligned. In **Fig. 2(b)**, a dog sound is played at the location of a cat, while the silent dog is visually present at a different location. Among all the image-audio pairs in Congruent condition, we randomly choose an object from a non-target category as the sound source. We do not limit the distance between the distractor and the target.

**Absent Visual Cue (AbsVcue)** explores the case when a target sound is present but the visual scene contains non-relevant objects, and no visual cue matches the sound. For instance, in **Fig. 2(c)**, a dog sound might be played on the cat, but no dog is visually present. From the image-audio pairs in the congruent condition, we randomly select a target sound to play at a randomly selected sounding object in the scene where no relevant objects aligning with the semantics of the sound source exists. This condition is more stringent than Conflicting Visual Cue, as it lacks any visual cues altogether.

**Audio Only (AOnly)** represents the extreme case where no meaningful visual information is provided, and the sound source is randomly placed anywhere within the image. The image could be a blank gray image with pixel values set to 128 (**Fig. 2(d)**) or a Gaussian noise image with a mean of 0 and a standard deviation of 1. For example, a dog sound could be randomly played on the left side of a pure gray background.

**Vision Only (VOnly)** exploits multi-modal biases or priors. In this condition, only visual scenes are provided to the AI models, and they must localize the sounding object despite the absence of any meaningful sound. The audio could either be completely silent or filled with random Gaussian noise (mean 0, standard deviation 1). For example, the same visual stimulus as in the congruent condition is presented, but the dog sound is replaced with silence or noise (**Fig. 2(e)**).

**Multi-Instance Localization (MultiInstLoc)** follows the same motivation as the cocktail party problem [29] and features several objects of the same category, but the audio corresponds to only one specific instance, testing localization accuracy in a multi-instance scenario. For example, as illustrated in **Fig. 2(f)**, a dog sound is played at the location of the left dog, while both dogs are visually present in the scene. This condition follows the same setup as the Congruent condition, but is more challenging due to the presence of multiple visually relevant objects. We selected images from the test set of the MSCOCO dataset, where the number of object instances within a given image is restricted to between 2 and 5 for the target category.

## 2.3 Human psychophysics experiment

We conducted in-lab psychophysics experiments on the AudioCOCO test set with 14 participants, collecting a total of 2,100 trials. All the experiments are conducted with the subjects' informed consent and according to protocols approved by the Institutional Review Board of our institution. Every experiment lasted approximately 40 minutes. The experimental setup is schematically illustrated in **Fig. 2(h)**. Each trial began with a fixation cross displayed for 500 milliseconds, followed by the presentation of an image and audio pair drawn from one of the six experimental conditions (see **Sec 2.2**). The 6-second audio clip paired with the image stimulus continuously loops until

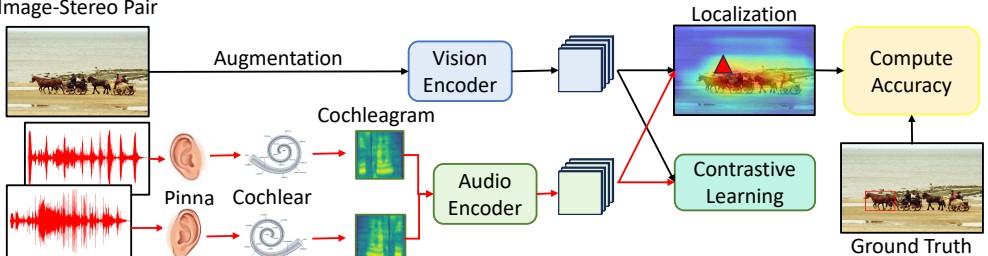

Figure 3: **Overview of our neuroscience-inspired EchoPin model.** EchoPin takes as input a static image paired with a two-channel stereo audio signal. The stereo waveforms are first filtered using the Head-Related Transfer Function (HRTF) to simulate sound filtering by the pinnae, and then converted into cochleagrams to mimic auditory processing in the human cochlea. The audio and visual streams are independently processed through dedicated encoders. During training, semantic alignment between the two modalities is enforced using a contrastive loss applied to paired audio and visual embeddings, while localization alignment is achieved by regressing the predicted sound source location (indicated by a red triangle) from the multimodal feature similarity map to the ground-truth location (red bounding box).

the trial concludes. Participants were instructed to use a computer mouse to click on the perceived location of the sound source within a time limit of 20 seconds, while wearing stereo headphones throughout the experiment. All trials were randomly sampled, and their presentation sequences were shuffled to minimize order effects. If participants failed to respond within the allotted 20 seconds, the trial automatically ended, and the next trial commenced. Instead of relying on the pre-rendered image-audio pairs, we conducted an audio calibration procedure at the start of the experiment to account for individual differences in auditory perception. To achieve this, we implemented real-time stereo audio synthesis in Unity. Following calibration, an audio validation task was conducted to ensure successful calibration and adequate spatial hearing accuracy from the participants. See **Supp Sec. S2 and Supp Fig.S6** for additional details on the human psychophysics experiments.

## 2.4 Our neuroscience-inspired EchoPin model

We introduce EchoPin, a neuroscience-inspired model for SSL (**Fig. 3**). The model takes image–audio pairs as input and emulates the human auditory periphery to decompose sounds in raw waveforms into frequency components. These representations are then aligned with visual features through a dual-encoder architecture for auditory and visual processing.

In human auditory neuroscience, incoming sound waveforms are directionally shaped by the pinna, head, and torso. These spectral transformations encode elevation-specific notches and interaural differences, which are essential for resolving front–back and vertical ambiguities [1]. For implementation, we use pre-measured Head-Related Transfer Functions (HRTFs) from human ears, developed in Unity simulations and based on the extensive KEMAR dummy head dataset [74]. KEMAR is equipped with microphones in the ear canals to capture how sounds from different directions are filtered by the head and pinnae. The dataset includes left and right ear impulse responses recorded from a Realistic Optimus Pro 7 loudspeaker positioned 1.4 meters from KEMAR, covering 710 spatial positions with elevations from $-40°$ to $+90°$.

Next, the HRTF-filtered time-domain sound waveform is passed through a cochlear-inspired frequency decomposition, converting it into a cochleagram using the PyCochleagram library [74]. The cochleagrams are constructed via Equivalent Rectangular Bandwidth (ERB) filterbanks, capturing the tonotopic and temporal resolution of sound across frequency channels. This representation retains key auditory features, including pitch, timbre, and spatial cues. The resulting 10-second stereo waveform, sampled at 16 kHz, is transformed into cochleagrams, yielding a tensor of size $66 \times 160{,}000 \times 2$, where the dimensions correspond to 66 ERB filters (after truncating 10 high-frequency channels for efficiency), 160k temporal samples, and two binaural channels. The binaural channels are first integrated using 1D convolution kernels to merge information across ears and then fed into the dual-encoder architecture of IS3 [45], allowing separate audio and visual processing streams.

During training, we initialize all weights from the pre-trained IS3 model, except for the first 1D-convolution layer in the audio encoder described above, and then optimize all parameters end-to-end using supervised learning. Two losses in the IS3 [45] model are employed: (i) a

Triplet loss to enforce semantic alignment by pulling matched audio–visual embeddings closer than mismatched ones, and (ii) a CIoU loss to penalize spatial deviation between predicted and ground-truth sounding-object bounding boxes. This combination enables EchoPin to jointly capture what is sounding and where it is located. See **Supp Sec. S3** for extra implementation details.

**Model variants of EchoPin.** To study the effects of design components in EchoPin, we introduce two model variants: EchoPin-M (Mono) averages the two HRTF-filtered stereo channels into a single time-domain waveform, allowing us to examine the impact of mono versus stereo audio on SSL tasks. EchoPin-S (Stereo) uses the HRTF-filtered stereo waveforms as input, but processes them with standard mel-spectrograms instead of cochleagrams. Comparing these variants with the full EchoPin model allows us to examine how stereo structure captured by the pinnae and frequency-specific cochlear modulation affect SSL performance. See **Tab. 1(b)** and **Sec. 3** for results and discussion.

## 2.5  Baseline methods and evaluation metrics

We benchmark EchoPin and the state-of-the-art multimodal models, including SSLTI [75], LVS [44], FNAC [43], CAVP [42], AVSegformer [23], IS3 [45], ImageBind [76], and LanguageBind [33], using the same stimuli as in our human psychophysics experiments. While humans can leverage the pinna, head, and torso to encode elevation-specific auditory cues, models lack these physical structures. To ensure fair comparisons across models and with human participants, all AudioCOCO audios are HRTF-filtered to approximate human auditory processing, and these filtered sounds are used for all model evaluations. In the main text, we provide brief overviews of IS3 [45] and a random baseline, and report their performance alongside our proposed EchoPin model. Detailed descriptions of the other models and their extended experimental results are provided in **Supp. Sec. S3**.

**IS3** [45] is a dual-stream architecture with 2D CNNs, which processes visual and monaural auditory inputs separately using dedicated encoders before fusing the features for contrastive learning during training. IS3 also includes an Intersection-over-Union (IoU) loss and a semantic alignment loss to improve localization accuracy during supervised training. The model is trained on the FlickrSoundNet [58] and VGG-Sound [49] datasets. **Random** is a chance model that randomly selects a location on the image as the predicted sound source location. It serves as a lower bound for SSL without using any audio-visual information.

**Predicting target sound locations.** For IS3, EchoPin, and other 2D CNN–based models, feature maps from the final layers of the visual and audio encoders are extracted, and cosine similarity is computed between them to generate a similarity heatmap. The predicted sound location is taken as the point with the highest activation on this heatmap. See **Supp. Fig. S7** for an illustration of how transformer-based baselines predict target sound source locations.

**Evaluation metrics.** To disentangle spatial localization from semantic alignment between visual and audio modalities, we define two metrics: **Audio Accuracy (A-Acc)** measures whether the model or human localizes the true sound source regardless of matching semantics. A-Acc = 1 if the peak activation falls within the bounding box of the sounding object; 0 otherwise. **Vision Accuracy (V-Acc)** measures alignment with visual semantics. V-Acc = 1 if the peak activation falls within any object of the correct category, even if it might not be the actual sound source, such as in MultiInstLoc conditions. In VOnly condition, V-Acc = 1 if the activation overlaps with any object from the 12 sound-emitting categories in AudioCOCO.

To robustly evaluate model performance, we consider three complementary factors that could influence results in **Supp Sec. S5**. First, A-Acc can be biased by object size, so we introduce a chance-corrected A-Acc (**Supp Tab. S5**; **Supp Tab. S1**). Second, human clicks may fall near but not inside the target, so we treat clicks within a thresholded radius as correct in **Supp. Tab. S2**. None of these metric variants alter the conclusions. Finally, we evaluate the predicted target object bounding boxes by all the models using corrected Intersection over Union (cIoU, [77]), where EchoPin continues to outperform other baselines (**Supp Sec. S5** and **Supp Tab. S3**).

## 3  Results

We report results from both human participants and AI models across all experimental conditions and object sizes. For brevity, the main text focuses on comparisons between humans and the two

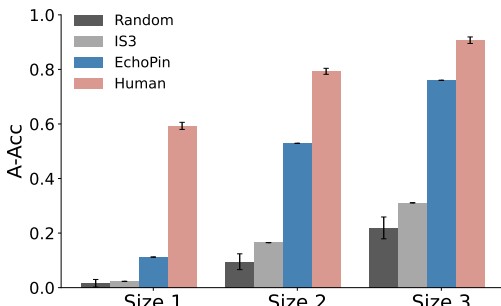

Figure 4: **Object size matters for humans and AI models in the congruent condition**. Accuracy increases with object sizes for both humans and AI models, with humans outperforming AI models, especially for small targets. Here and in subsequent figures, error bars represent Standard Error Mean (SEM).

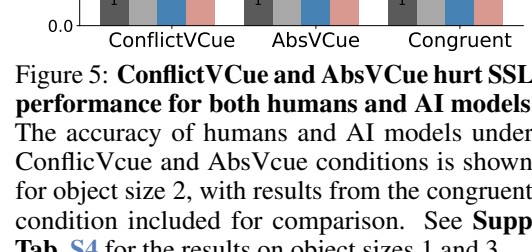

Figure 5: **ConflictVCue and AbsVCue hurt SSL performance for both humans and AI models**. The accuracy of humans and AI models under ConflicVcue and AbsVcue conditions is shown for object size 2, with results from the congruent condition included for comparison. See **Supp. Tab. S4** for the results on object sizes 1 and 3.

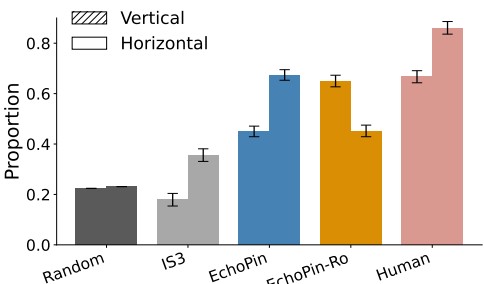

Figure 6: **EchoPin shows human-like horizontal-vertical asymmetry in SSL accuracy, while other models do not.** We report the proportion of trials (object size 2) where the predicted sound location falls within 6 degrees of visual angle from the ground truth, separately along the horizontal (textured bars) and vertical (plain bars) directions.

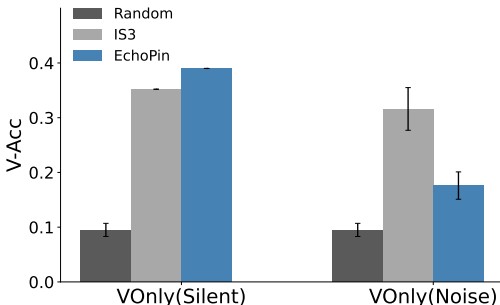

Figure 7: **AI models show a bias towards objects that emit sounds, even in the absence of sound or in the presence of noisy sound.** The V-Acc of AI models is presented in the VOnly condition, where either no sound or only noisy sound is present. Despite the lack of auditory cues, V-Acc of AI models on object size 2 remains higher than random (dark gray).

best-performing models, IS3 and EchoPin. Additional qualitative analyses for all other models are presented in **Supp. Tab. S4**.

**Object size matters for humans and AI.** As illustrated in **Fig. 4**, human A-Acc steadily increases with object size, suggesting that larger sounding objects are easier for humans to localize. EchoPin exhibits a similar trend, with A-Acc rising substantially from 11.2% to 76.0% as object size grows. Across all object sizes, EchoPin consistently outperforms IS3 and the other models. This difference likely arises because IS3 is trained on datasets biased toward large, centered objects, whereas AudioCOCO provides a wider range of object sizes and spatial locations. While both humans and AI models perform above chance for large objects, only humans and EchoPin maintain strong, consistent performance across all object sizes, including smaller ones—performance that IS3 fails to achieve.

**Conflict cues harm more than the lack of cues for AI.** As shown in Fig. 5, both the ConflictVCue and AbsVCue conditions impair SSL performance in humans compared to the congruent conditions. While humans and EchoPin still perform significantly above chance, IS3 drops to near-chance levels under the ConflictVCue condition. This suggests that humans and EchoPin are more robust in SSL tasks involving conflicting or missing visual cues, whereas IS3 relies heavily on visual information. Notably, unlike humans, EchoPin shows a greater performance decline when visual cues are conflicting but not when they are absent, indicating that it is more easily misled by incongruent visual information yet remains stable in the absence of such cues.

**AI fails when there are only auditory cues but humans can.** As shown in **Supp. Fig. S8**, humans achieve above-chance A-Acc even without visual information (e.g., gray or Gaussian noise

| (a) Multi-Instance Localization | | | | | | | | (b) Overall Performance | | | | |
|---|---|---|---|---|---|---|---|---|---|---|---|---|
| Acc(%) | | Rand | IS3 | CAVP | AVSeg | EchoPin | Human | Acc(%) | | Rand | IS3 | EchoPin-M/S | EchoPin |
| A-Acc | Size1 | 1.6 | 4.8 | 2.9 | 2.5 | 4.5 | **25.7** | Mono | Size1 | 1.6 | 3.0 | 3.6 | - |
| | Size2 | 9.1 | 7.9 | 7.5 | 7.3 | 24.1 | **36.4** | | Size2 | 9.4 | 13.9 | 15.8 | - |
| | Size3 | 21.3 | 22.4 | 20.4 | 20.2 | 47.1 | **38.6** | | Size3 | 19.8 | 28.7 | 31.4 | - |
| V-Acc | Size1 | 8.4 | 11.9 | 10.5 | 11.2 | 37.5 | **60.9** | Stereo | Size1 | 1.6 | - | 5.3 | **9.7** |
| | Size2 | 17.8 | 24.1 | 23.0 | 23.7 | 53.8 | **82.8** | | Size2 | 9.4 | - | 17.0 | **31.3** |
| | Size3 | 26.2 | 40.9 | 39.5 | 40.9 | 64.2 | **89.1** | | Size3 | 19.8 | - | 35.2 | **47.6** |

Table 1: **Multi-instance SSL remains a challenging task for both humans and AI models.** The table (a) on the left summarizes the audio and visual localization accuracy of humans and AI models under the multi-instance condition across all object sizes. **For AI models, both input data quality and the use of stereo audio substantially impact SSL performance.** Table (b) on the right summarizes the average A-Acc across the Congruent, ConflictVcue, AbsVcue, and AOnly conditions for models trained with either mono or stereo audio. The second-to-last column shows the results of EchoPin-M (Rows 1–3) and EchoPin-S (Rows 4–6). See **Sec. 2.4** for details on these variants. (–) indicates that the results are not applicable due to the model configurations. In both tables, best is in bold and the second best is underlined.

backgrounds), confirming that they can perform SSL based solely on auditory cues. However, their performance remains lower than in the congruent condition, indicating that congruent visual information facilitates SSL. Similar trends are observed in IS3 and EchoPin, with EchoPin consistently outperforming IS3 under both AOnly conditions (gray or Gaussian background). Interestingly, both humans and EchoPin perform better with gray than Gaussian backgrounds, suggesting that incongruent visual noise can distract attention and interfere with SSL, whereas a neutral (gray) background minimizes interference.

**EchoPin shows human-like asymmetry in auditory spatial precision.** In neuroscience, auditory spatial precision is known to exhibit a horizontal–vertical asymmetry: humans localize sounds more accurately along the azimuth (horizontal) than the elevation (vertical) axis [71]. To quantify this effect, we measured the proportion of trials where predictions fell within six degrees of visual angle from ground-truth locations, separately for horizontal and vertical dimensions (**Fig. 6**). As expected, humans showed a strong horizontal advantage, localizing targets in 86.1% of trials horizontally but only 66.7% vertically. Remarkably, EchoPin exhibited a similar asymmetry pattern despite being trained without any human behavioral data. In contrast, IS3, which relies solely on monaural Mel-spectrograms, also showed asymmetry but to a much lesser degree. Although both models have benefited from spatial filtering by the pinnae, the observed asymmetry in EchoPin arises from its biologically grounded auditory frequency decomposition in the cochlea and its stereo audio perception. To further validate this, we introduced an EchoPin variant, EchoPin-Ro, in which the interaural axis was rotated by 90 degrees in Unity, effectively simulating vertically aligned ears. When audio was re-rendered under this configuration, the model's asymmetry was reversed, confirming the structural origin of this effect.

**AI biases toward sound-emitting objects.** Previous studies [42, 45] show that AI models often exploit visual shortcuts by favoring large or centered objects. To examine additional behavioral biases, we evaluate models under the VOnly condition. Without meaningful sound, any object could plausibly be the sound source. As shown in **Fig. 7**, models tend to localize sounds to vocal objects (e.g., people, animals) rather than irrelevant ones (e.g., sky, trees), resulting in above-chance V-Acc. This indicates that models encode prior knowledge of which objects typically produce sound. Furthermore, for all the models, Gaussian noise audio input leads to lower V-Acc than the absence of audio, indicating that even noisy audio can reduce the models' over-reliance on visual signals.

**Multi-instance SSL remains challenging for humans and AI.** As shown in **Tab. 1a**, V-Acc is high for both EchoPin and humans, indicating strong alignment between audio and visual semantics. This allows them to use audio cues to identify all visual objects with matching semantics. However, in scenes with multiple object instances, successful localization requires fine-grained SSL, beyond semantic matching. In these cases, A-Acc drops for both humans and EchoPin. Compared to the Congruent condition (with a single sounding object), multi-instance SSL is notably harder, as it demands precise spatial disambiguation. Despite this, humans still perform above chance, especially for small objects. Similarly, EchoPin achieves high A-Acc on object sizes 2 and 3. However, it still

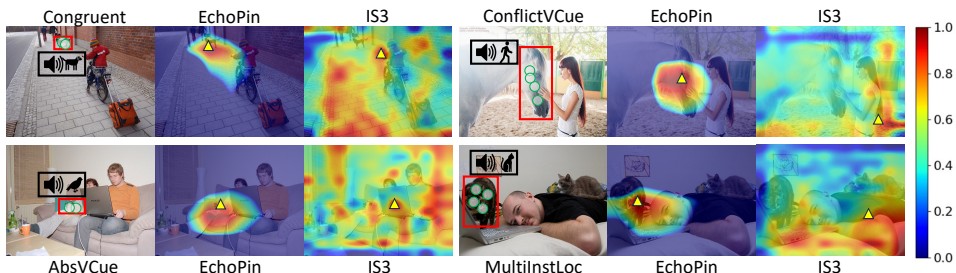

Figure 8: **IS3 struggles to localize sound sources, whereas humans and EchoPin perform well across all four experimental conditions.** The leftmost images (Columns 1 and 4) show the correct localization results made by human participants. Red boxes mark the ground truth sound source locations, while green circles indicate mouse click responses from multiple participants. The middle columns (2 and 5) and rightmost columns (3 and 6) display heatmaps predicted by EchoPin and IS3, respectively. Yellow triangles on the heatmaps denote the predicted sound source locations from each model. See the colorbar for the activation values of the heatmaps.

lags behind human performance, with the gap more pronounced in small object size. Despite this, both EchoPin and humans still perform above chance, especially for small objects. This demonstrates an ability to localize sounds at fine spatial resolution for both humans and EchoPin. Among all models, EchoPin performs best. It even outperforms strong baselines like CAVP and AVSeg, which are trained on large-scale, standard audio-visual datasets.

**Training data quality and stereo input are important for SSL.** We report the average A-Acc across all object sizes for Random, IS3, and the EchoPin variants under four experimental conditions in **Tab. 1**b. EchoPin consistently achieves the highest performance across all conditions. Notably, **EchoPin-M** surpasses IS3 despite using fewer fine-tuning examples, underscoring the importance of high-quality training data. Moreover, **EchoPin-S** further improves over **EchoPin-M**, highlighting the advantage of incorporating human-like stereo configurations for spatial localization.

We further visualize the predicted sound source locations from human participants, IS3, and EchoPin in **Fig. 8**. EchoPin localizes sound sources more accurately and often aligns closely with human judgments. In contrast, IS3 struggles with small or peripheral targets—for instance, it fails to localize a dog in the top-left corner under the Congruent condition and frequently misattributes sounds to other vocal objects (e.g., person, motorbike). Although EchoPin markedly improves SSL robustness, it is still inferior to human performance in complex scenes. Failure cases and additional comparisons with other baselines, such as CAVP, are provided in **Supp. Sec. S4**, **Supp. Fig. S9**, and **Supp. Fig. S10**.

## 4 Discussion

We systematically and quantitatively examine modality biases and conflicts in SSL across humans and AI models. Our study covers six audiovisual conditions, including congruent cues, conflicting signals, and cases with missing audio or visual input in natural scenes. Human listeners show strong robustness: although conflicting or absent visual cues reduce performance, they can still accurately localize even small sound sources under challenging or multi-instance conditions, and even in the absence of visual input. In contrast, current multimodal AI models rely heavily on vision—misattributing sounds to large, centered, and salient objects, and suffering steep performance drops when visual cues are removed. Conflicting visuals further degrade their accuracy, with most models performing near chance for small or visually absent sound sources.

We identify two primary causes of these limitations: (1) low-quality, visually biased audiovisual datasets, and (2) monaural audio inputs lacking spatial fidelity. To overcome these issues, we curate AudioCOCO, a high-quality dataset built through rigorous filtering and 3D physical simulation. By integrating depth maps, HRTF-based filtering that mimics pinna effects, cochlear-inspired frequency decomposition and modulation, and physically grounded 3D sound propagation, AudioCOCO produces realistic, spatialized stereo audio aligned with human auditory processing. Building on this, we introduce EchoPin, a neuroscience-inspired model trained on AudioCOCO. Despite fewer training examples, EchoPin surpasses state-of-the-art models across all conditions and exhibits human-like localization biases, such as stronger precision along the horizontal plane. Ablation studies confirm the importance of both high-quality datasets and stereo auditory input for capturing spatial cues. This work underscores the value of designing models and datasets that respect the physical constraints of

sensory systems. Future directions include scaling AudioCOCO to incorporate temporal dynamics from videos and improving realism in simulated sound rendering, such as the effect of refraction.

## Acknowledgements

This research is supported by the National Research Foundation, Singapore under its NRFF award NRF-NRFF15-2023-0001 and Mengmi Zhang's Startup Grant from Nanyang Technological University, Singapore. We would also like to thank Qing Lin and Shuangpeng Han for their valuable advice and feedback on the project.

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

# Supplementary Material

In the supplementary material, we provide additional information of our AudioCOCO dataset in **Sec.S1** and human experiment setup in **Sec.S2**. Moreover, we include extra experimental results, an ablation study, and visualization results for predicted position bias, which are discussed in **Sec.S4** and **Sec.S5**.

## S1    Details about the AudioCOCO dataset

**More details on image selection.** The MSCOCO 2014 dataset [47] offers annotations for 80 object categories, including both bounding boxes and segmentation masks. To ensure a fair comparison—especially given that some model backbones are pre-trained on MSCOCO—we utilized the MSCOCO 2014 validation set rather than the training set. From this, we selected 12 common vocal categories frequently encountered in daily life: person, motorbike, train, boat, elephant, bird, cat, dog, horse, sheep, cow, and keyboard. A total of 29,737 images containing at least one vocal object were extracted to form the pool of image candidates for our dataset.

To investigate spatial distribution, we visualized the occurrence frequency of these categories across the 29,737 images in **Fig. S1b** and **Fig. S1a**, grouping instances by three object area size bins defined in the main draft: object size1 (0-5%), size2 (5-15%), and size3 (15-30%). For smaller object sizes, category locations were relatively uniformly distributed across the image space. However, for larger objects, some categories—such as bus, train, and truck—exhibited a strong center bias. To correct for this, we filtered the dataset to ensure that no position in the final heatmap exceeded 50% in frequency.

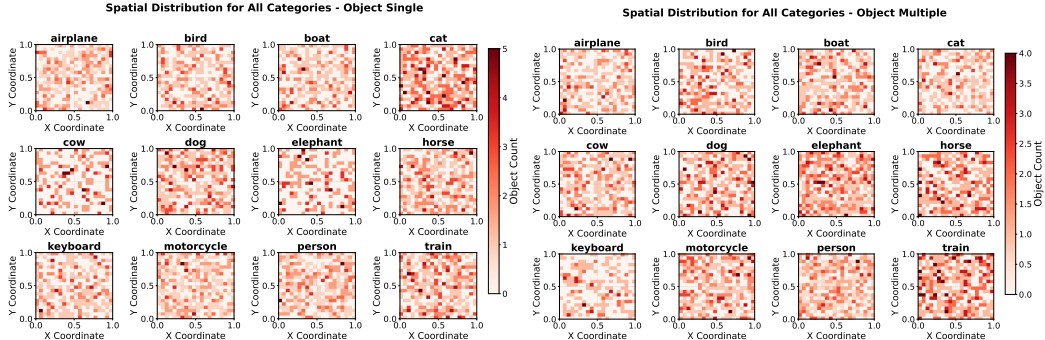

(a) The spatial distribution of single objects by category. (b) The spatial distribution of multi-objects by category.

Figure S1: The visualization results of images' spatial distributions in the AudioCOCO test set are presented for each object category, where the color bar indicates the frequency of object occurrences at each spatial location within the images. Darker regions correspond to higher frequencies.

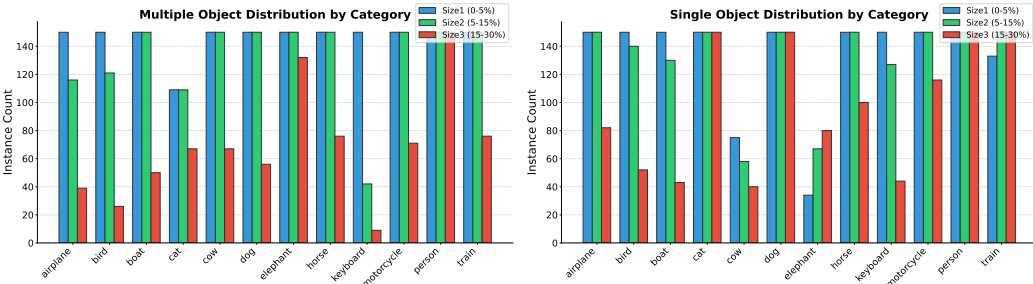

(a) The count distribution of multi-objects by category. (b) The count distribution of single objects by category.

Figure S2: The visualization results of the image count distributions in the AudioCOCO test set are presented, where blue, green, and red represent object size1, size2, and size3, respectively. This visualization illustrates how instances are distributed across different object size categories, highlighting the dataset's balanced distribution in terms of object sizes.

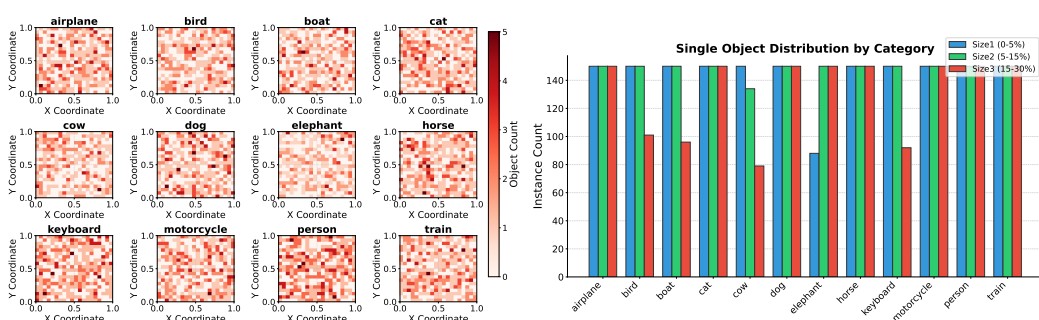

(a) The spatial distribution of single objects by category for AudioCOCO's training set.

(b) The count distribution of single objects by category for AudioCOCO's training set.

Figure S3: The visualization results of the image count distribution and spatial distribution for the AudioCOCO training set. For the count distribution, object sizes are color-coded: blue for Size1, green for Size2, and red for Size3. This plot illustrates the number of object instances across different size categories, confirming that the dataset maintains a balanced distribution in terms of object size. For the spatial distribution, the heatmap reflects the frequency of object occurrences at each spatial location in the images, with the color bar indicating frequency. Darker regions correspond to areas of higher object density, highlighting positional biases or spread across the dataset.

Additionally, given that the image distribution over all object categories in COCO is not uniform, we performed random sampling to cap the number of instances to 150 per category per object size bin, promoting balanced representation within the AudioCOCO dataset. The final distributions after filtering are shown in **Fig. S2a**, **Fig. S2b**, and **Fig. S3**, demonstrating that AudioCOCO achieves a balanced dataset across object area sizes, object center positions, and categories.

**More details on audio selection.** To ensure that only high-quality, semantically aligned clips are retained, we apply the following three filtering steps to the videos from VGGSound [49]. First, we introduce Semantic Consistency (SeC) and select audios that are representative of the semantic object categories. Specifically, we take all audio files belonging to the same semantic category, extract audio features from the last layer of the Wav2vec [78] model, compute their pari-wise cosine similarities based on these features, and retain the top 80% audios with the highest cosine similarity scores. Second, to ensure the audio is free from noise or interference from unrelated sources, we introduce Mel-Spectrogram Similarity (MSS) as a filtering criterion within each category. For each audio clip, we compute its Mel spectrogram, average the frequency magnitudes over time, and apply a logarithmic transformation to compress high-frequency components—yielding a compact representation of the audio's overall spectral structure. We then calculate the cosine similarity between these representations and retain the top 65% of audio clips based on this MSS metric, following the SeC criteria. Third, to ensure the stereo audio corresponds to the sounding object being centered in the video frame, we introduce Spatial Consistency (SpC) for each audio-image pair. We compute the Spearman correlation between the left and right audio channels; a high correlation suggests the sound source is centrally located, producing similar waveforms in both channels. We retain the top 50% of audio clips based on this metric from the MSS-filtered set.

For audio selection, we also enforce a minimum threshold of 20 audio clips per category. If any filtering step results in a category falling below this threshold, that specific step is discarded, and the previous filtering output is retained as the final result to ensure sufficient data coverage across all categories. The final distributions after filtering are shown in **Fig. S4**.

## S2 Details of human psychophysics experiments

All the experiments are conducted with the subjects' informed consent and according to protocols approved by the Institutional Review Board of our institution. Participants were instructed to use a computer mouse to click on the perceived location of the sound source within a time limit of 20 seconds, while wearing stereo headphones (SENNHEISER MOMENTUM 4 with active noise cancellation) throughout the experiment.

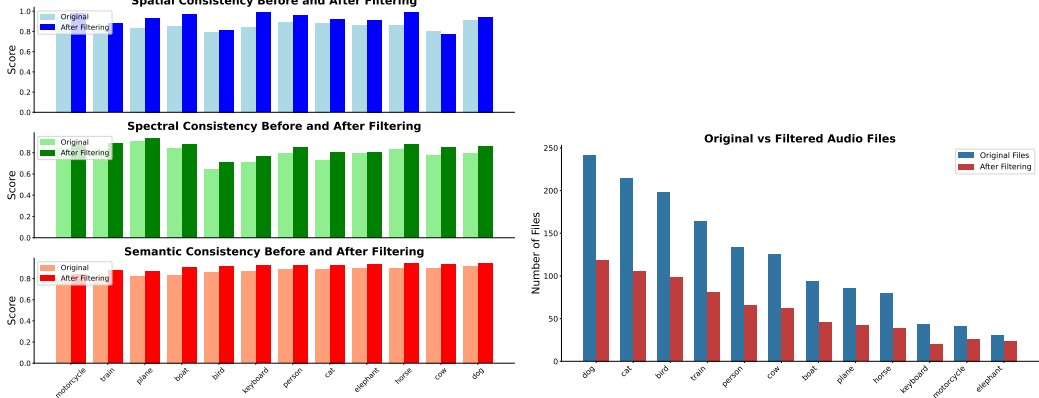

(a) The three stages of audio filtering result for AudioCOCO test set.

(b) The comparison between original audio counts and filtering audio counts for AudioCOCO test set.

Figure S4: The visualization results of audio statistics in the AudioCOCO test set demonstrate that the audio quality across most categories has significantly improved following the filtering process. Additionally, we ensured a balanced count distribution among all categories. These outcomes highlight the effectiveness of our selection and refinement strategy in enhancing both the semantic consistency and acoustic quality of the dataset.

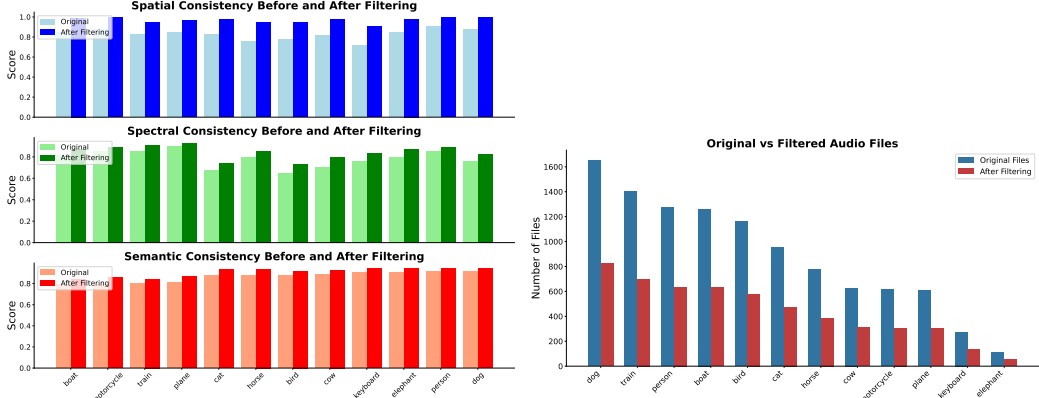

(a) The three stages of audio filtering result for AudioCOCO training set.

(b) The comparison between original audio counts and filtering audio counts for AudioCOCO training set.

Figure S5: The visualization results of audio statistics in the AudioCOCO training set demonstrate that the audio quality across most categories has significantly improved following the filtering process. Additionally, we ensured a balanced count distribution among all categories. These outcomes highlight the effectiveness of our selection and refinement strategy in enhancing both the semantic consistency and acoustic quality of the dataset.

**Audio calibration.** Instead of relying on the pre-rendered image-audio pairs, we conducted an audio calibration procedure at the start of the experiment to account for individual differences in height and auditory perception. To achieve this, we implemented real-time stereo audio synthesis in Unity, which communicates with MATLAB (hosting the human behavioral experiment) via TCP connections. The measured delay for real-time stereo sound synthesis and presentation is within 500 milliseconds, ensuring seamless interaction between the two systems.

During calibration, participants were presented with a white dot at a random location on the screen, accompanied by an audio clip spatially rendered at the dot's position. A white cross at the center of the screen served as a spatial reference (see **Supp Fig.S6**). Using the keyboard's arrow keys, participants adjusted the perceived audio source position until the sound aligned with the white dot.

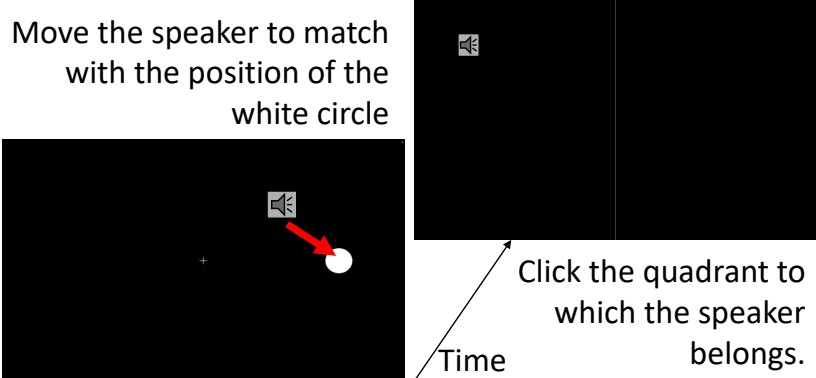

Figure S6: **The audio calibration and validation process**. During calibration, the player is at a random position close to the white dot. The red arrow represents a potential movement trajectory. During validation, the display is divided into two equal quadrants, with the audio player appearing at a random position three times within each quadrant.

Once satisfied, participants pressed the ESC key to confirm the alignment. This process was repeated six times with different white dot positions.

For each calibration step $t \in 1, 2, \ldots, 6$, we recorded the participants' adjustments of the sound source location in pixels $(\Delta x_t, \Delta z_t)$. The calibration hyperparameters $\alpha$ and $\beta$ were computed as the mean of $\Delta x_t$ and the sum of $\Delta z_t$, respectively. These hyperparameters were then applied to scale the Unity coordinates $x_u$ and $z_u$, correcting for individual perceptual biases in auditory localization during the main experiment. The rationale for using the mean adjustment for $\Delta x_t$ (horizontal direction) and the sum for $\Delta z_t$ (vertical direction) is based on human spatial hearing characteristics—listeners typically localize horizontal (azimuth) sound sources with higher precision than vertical ones. This design allows greater tolerance for variability in the vertical dimension (altitude) during calibration, ensuring more robust alignment with participants' perceptual expectations.

**Audio validation.** Following calibration, an audio validation task was conducted to ensure successful calibration and adequate spatial hearing accuracy. Participants heard an audio clip played from one of two possible locations on a horizontally arranged 1x2 grid displayed on the screen (see **Supp Fig. S6**). They were instructed to click on the half of the screen from which they perceived the sound. This validation procedure was repeated six times. If a participant's spatial sound localization accuracy fell below 83%, the calibration and validation procedures were repeated to guarantee reliable data quality during the actual experiment.

**Center fixation presentation before the visual stimulus onset.** During the experiment, participants were instructed to fixate on a central dot before each trial began. This pre-trial fixation is a standard element in human psychophysics and cognitive neuroscience, designed to recenter attention and minimize carry-over effects across trials. By requiring participants to begin each trial from a common spatial and attentional baseline, this design ensures that any differences in response latency or eye movement patterns can be attributed to the experimental manipulation, rather than lingering attentional bias from the previous trial.

## S3    More implementation details of AI models

**SSLTI [75], LVS [44], FNAC [43], and IS3 [45]** are SSL models based on dual-stream architectures with 2D Convolutional Neural Networks (CNNs). Each model processes visual and auditory inputs separately using dedicated encoders before fusing the features for contrastive learning during training. Both encoders are based on ResNet18 [79]. Beyond standard contrastive learning, IS3 introduces an Intersection-over-Union (IoU) loss and a semantic alignment loss to improve localization accuracy and better alignment between audio and visual modalities during supervised training. All the models are trained on the FlickrSoundNet [58] and VGG-Sound [49] datasets.

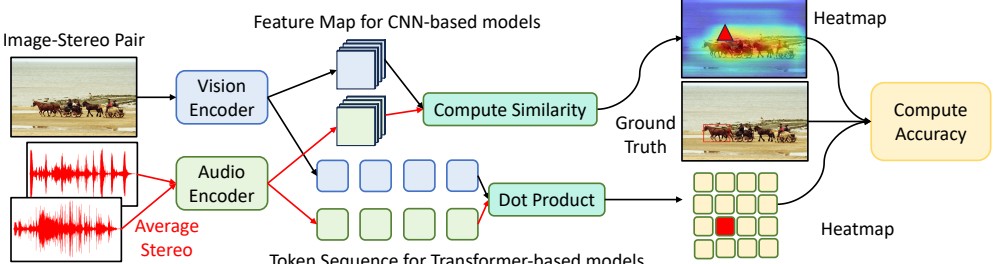

Figure S7: **Overview of SOTA multi-modal models for sound source localization (SSL).** All models receive paired images and mono audio inputs, where stereo signals are averaged into a single channel. Visual and auditory inputs are processed independently through separate encoders. For CNN-based models (indicated by blue arrows), feature maps from the final layers of each encoder are extracted, compared via cosine similarity, and used to generate a similarity heatmap. For transformer-based models, output token sequences are obtained from both encoders. Dot products on their token embeddings are calculated, and a heatmap is produced accordingly. During evaluation, SSL accuracy is determined by verifying whether the location of the maximum activation on the heatmaps (red triangles or red tokens) lies within the segmentation mask of the ground truth sounding object (red bounding box).

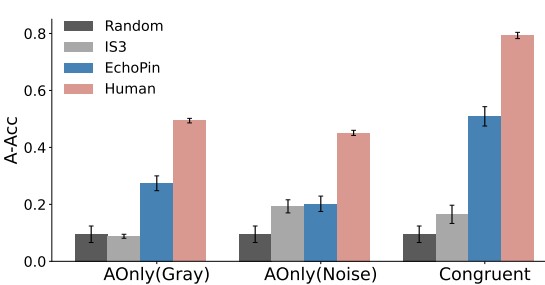

Figure S8: **While humans benefit from visual cues, they remain capable without them in the AOnly condition, unlike AI models, which fail to localize sound sources without congruent visual information.** The accuracy under AOnly conditions with both grayscale and Gaussian noise backgrounds drops for both humans and AI models compared to the congruent condition on object size 2.

**CAVP [42]** is a sound source segmentation model that follows a dual-stream CNN architecture. It uses PVTV2-B5 [80] as the visual encoder and either VGGish [81] or ResNet18 as the audio encoder. CAVP is trained in a fully supervised manner using cross-entropy loss for segmentation and contrastive learning to align audio-visual features. Training data includes AVSBench [35] and VPO [42] datasets.

**AVSegformer [23]** is an audio-visual semantic segmentation model built on a dual-stream transformer-based architecture. It employs SegFormer [82] as the visual backbone and Audio Spectrogram Transformer (AST) [83] as the audio encoder to capture fine-grained cross-modal representations. AVSegformer integrates both modality-specific and fused token embeddings through a lightweight fusion decoder for pixel-level prediction. It is trained in a fully supervised setting using a combination of cross-entropy loss for segmentation and audio-visual consistency loss. AVSegformer's training data includes three subsets of AVSBench.

**ImageBind [76] and LanguageBind [33]** are large-scale, multi-modal transformer models with billions of parameters. These models are trained with contrastive learning objectives across six modalities: image, audio, video, text, depth, thermal, and IMU, and embed these into a shared representation space. Both models use a Vision Transformer(ViT) [84] for visual encoding and AST [83] for audio encoding. ImageBind is trained on large-scale datasets including LAION [85, 86], SSv2 [87], and K400 [88]. LanguageBind builds on the pre-trained encoders from ImageBind and further fine-tunes them on the VIDAL-10M [33] dataset.

**Implementation details of AI models.** All eleven models—except EchoPin-S and EchoPin—take paired image and mono audio inputs, processing each modality independently through dedicated visual and audio encoders. For the mono audio setup, we follow the EchoPin-M design by averaging the stereo channels into a single-channel input. As shown in Fig. S7, for CNN-based models, feature

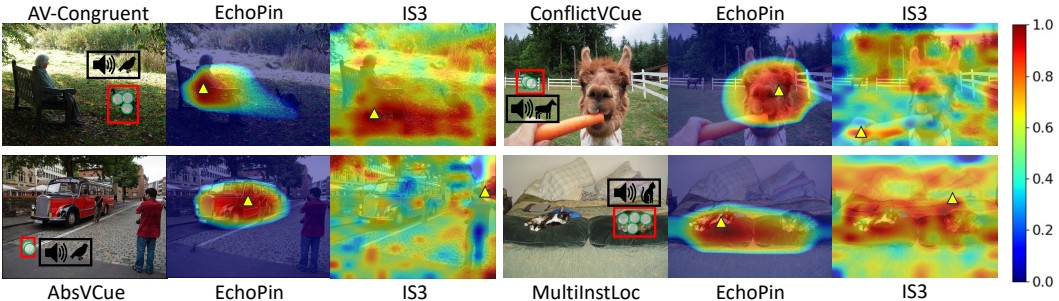

Figure S9: The figure shows the failure examples of both IS3 and EchoPin compared to humans. The leftmost images (Columns 1 and 4) show the correct localization results made by humans. Red boxes mark the ground truth sound source locations, while green circles indicate mouse click responses from multiple participants. The middle columns (2 and 5) and rightmost columns (3 and 6) display heatmaps predicted by EchoPin and IS3, respectively. Yellow triangles on the heatmaps denote the predicted sound source locations from each model. See the colorbar for the activation values of the heatmaps.

maps from the final layers of the encoders are extracted, and cosine similarity is computed between them to produce a similarity heatmap. For transformer-based models, token sequences from both encoders are retrieved, and pairwise dot products of their token embeddings are calculated to generate the heatmap. The predicted sound source location is identified as the point with the highest activation on the heatmap.

We evaluated all the current models using their publicly available, pre-trained weights and adhered to their original implementation details. Since AVSBench [35] shares some images with our AudioCOCO test set, we exclude those overlapping image-audio pairs when evaluating CAVP [42] and AVSegformer [23]. Experiments for ImageBind and LanguageBind were conducted on 8 NVIDIA A100 GPUs, whereas all other models, including EchoPin and its variants, were trained and evaluated on 4 NVIDIA A6000 GPUs. We fine-tune the EchoPin models using the Adaptive Moment Estimation (Adam) optimizer [89] with a weight decay of $1 \times 10^{-4}$ for 10 epochs. The initial learning rate is set to $1 \times 10^{-5}$, and the batch size is 16. Each fine-tuning session takes approximately 16 hours to complete. To accelerate data loading, all audio waveforms are preprocessed and stored as cochleagram tensors in advance, with each .npy file occupying roughly 160 MB. All model evaluations are repeated three times with different random seeds to ensure statistical reliability.

## S4    More qualitative results of humans and AI in SSL

As shown in **Fig.S9** and **Fig.S10**, we further illustrate the limitations of EchoPin relative to human performance and provide additional qualitative results for other baselines. Notably, EchoPin often fails to localize the correct sounding object when a visually salient distractor occupies a large portion of the scene. This suggests that the model remains vulnerable to visual saliency bias, tending to prioritize large or central objects even when they are not the true auditory source—a tendency that human listeners are better equipped to suppress.

Similar patterns are shown in CAVP, which exhibits modality conflict sensitivity akin to IS3. Both models are frequently misled by conflicting audio-visual cues and demonstrate a systematic bias toward the visual modality, highlighting a lack of robust auditory grounding under incongruent conditions.

## S5    More quantitative results of AI models in SSL

The raw A-Acc may be confounded by the area of the ground-truth mask—since, intuitively, smaller objects are more difficult to localize while larger ones are easier. To address this potential bias, we introduce a chance-corrected gain metric, which quantifies the improvement of a human or model over a random guess, normalized by the baseline accuracy of the random guess:

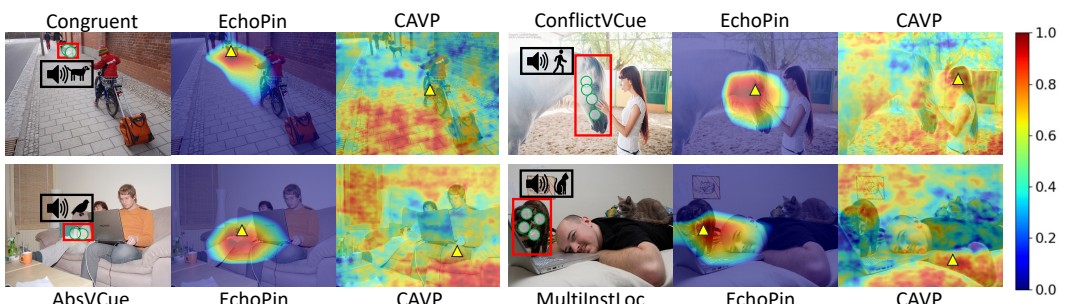

Figure S10: The figure shows the failure examples of CAVP and EchoPin compared to humans. The rightmost columns (3 and 6) display heatmaps predicted by CAVP.

$$\text{Gain}(X) = \frac{\text{Acc}_X - \text{Acc}_{\text{rand}}}{\text{Acc}_{\text{rand}}} \times 100 \qquad (1)$$

The $\text{Acc}_{\text{rand}}$ represents the accuracy achieved when predictions are sampled uniformly at random across the scene.

| Normalized Percentage | Size 1 | Size 2 | Size 3 |
|---|---|---|---|
| (EchoPin - Random) / Random | 5.2 | 4.6 | 2.9 |
| (Human - Random) / Random | 31.9 | 7.3 | 3.6 |
| (Human - EchoPin) / Random | 26.7 | 3.0 | 0.8 |

Table S1: We report the chance-corrected gain for humans and models across three object sizes under the congruent condition. While human performance increases significantly with larger object sizes, models show limited improvement. This discrepancy may stem from AudioCOCO's uniform object size distribution, which limits the models' ability to exploit size-dependent cues.

This normalization removes the influence of mask size and isolates the true localization capability of humans and models. As shown in Tab.S1, even after correcting for chance, humans consistently outperform models—especially for small object sizes. For example, in the smallest size category, humans achieve a gain of 31.9%, compared to only 5.2% for models. These results suggest that object size modulates sound source localization performance in a way that cannot be fully explained by ground-truth mask area alone, highlighting deeper perceptual and representational differences between human and model behavior.

We apply the same evaluation criterion to both human participants and AI models to ensure fair comparisons: a prediction is considered correct if the peak activation (for models) or the human click falls within the bounding box of the sounding object. We also conducted an additional analysis by varying the pixel distance thresholds used to determine correctness. Specifically, a prediction is considered correct if it falls within $x$ pixels of the ground-truth bounding box. We report the resulting A-Acc (accuracy with spatial tolerance) as a function of pixel thresholds in Tab.S2 based on congruent conditions for object size2. From these results, we observe that while larger thresholds naturally lead to higher A-Acc values, the relative performance trend between humans and models remains consistent. This further supports the validity of our evaluation methodology.

Moreover, we conducted an additional experiment by fine-tuning our EchoPin-S model using the VGG-SS and Flickr-SoundNet datasets, and report comparative results against IS3 and CAVP using the CIoU metric under the default evaluation protocol provided by [90]. As a localization-aware metric, CIoU accounts for overlap area, center distance, and aspect ratio alignment, offering a more comprehensive assessment of spatial prediction quality. From Tab. S3, we observe that benefiting from the fine-grained spatial features provided by AudioCOCO, our EchoPin-S model achieves superior performance on these standard SSL benchmarks, outperforming the baselines.

| Threshold | Random | Human | EchoPin-S | EchoPin |
|-----------|--------|-------|-----------|---------|
| 0 (default) | 9.5 | 79.3 | 17.3 | 50.9 |
| 10 | 9.6 | 79.9 | 18.5 | 52.0 |
| 25 | 9.9 | 80.4 | 19.8 | 52.4 |

Table S2: We compare performance across varying localization thresholds (0–25 pixels) under the congruent condition for object size 2. As shown, both human and model accuracy exhibit only marginal improvement with increasing thresholds, indicating that our evaluation metric is stable and not overly sensitive to small shifts in the decision boundary—thereby validating its robustness.

| Method | VGG-SS | Flickr-SoundNet |
|--------|--------|-----------------|
| IS3 | 42.96 | 84.40 |
| CAVP | 43.58 | 85.03 |
| EchoPin-S | 43.61 | 85.25 |
| **EchoPin** | **45.02** | **85.87** |

Table S3: We conduct comparison experiments on the VGG-SS and Flickr-SoundNet datasets. Despite the challenges posed by these large-scale, imbalanced public benchmarks, EchoPin consistently outperforms prior state-of-the-art (SOTA) methods, highlighting the robustness of spatial cues and the effectiveness of our neuroscience-inspired design.

| Model Acc(%) | Congruent | | | Conflicting VCue | | | Absent VCue | | | Audio Only | | | | | | Vision Only | | | | | | Multi-Instance | | | | | |
|---|---|---|---|---|---|---|---|---|---|---|---|---|---|---|---|---|---|---|---|---|---|---|---|---|---|---|---|
| | A-Acc | | | | | | | | | Vision Gray | | | Vision Noise | | | Audio Silent | | | Audio Noise | | | A-Acc | | | V-Acc | | |
| | Size1 | Size2 | Size3 | Size1 | Size2 | Size3 | Size1 | Size2 | Size3 | Size1 | Size2 | Size3 | Size1 | Size2 | Size3 | Size1 | Size2 | Size3 | Size1 | Size2 | Size3 | Size1 | Size2 | Size3 | Size1 | Size2 | Size3 |
| Random | 1.8 | 9.5 | 19.6 | 1.8 | 9.5 | 19.6 | 1.8 | 9.5 | 19.6 | 1.8 | 9.5 | 19.6 | 1.8 | 9.5 | 19.6 | 1.8 | 9.5 | 19.6 | 1.8 | 9.5 | 19.6 | 1.6 | 9.1 | 21.3 | 8.4 | 17.8 | 26.2 |
| IS3 | 2.3 | 16.5 | 31.1 | 1.7 | 12.3 | 24.8 | 3.3 | 13.5 | 29.8 | 2.3 | 8.8 | 11.9 | 3.3 | 17.3 | 39.7 | 21.5 | 35.2 | 42.7 | **20.5** | **31.6** | **38.9** | 4.8 | 7.9 | 22.4 | 11.9 | 24.1 | 40.9 |
| FNAC | 1.8 | 12.5 | 26.9 | 1.9 | 7.2 | 13.3 | 5.8 | 10.5 | 18.6 | 1.1 | 4.8 | 8.7 | 1.3 | 5.0 | 9.6 | 6.2 | 16.5 | 43.8 | 5.4 | 12.6 | 11.9 | 0.7 | 3.9 | 26.3 | 13.4 | 25.5 | 52.8 |
| SSLTIE | 1.4 | 9.9 | 23.7 | 2.1 | 6.8 | 12.4 | 5.2 | 9.9 | 16.5 | 0.7 | 3.9 | 8.2 | 1.1 | 4.3 | 8.9 | 6.1 | 16.5 | 40.7 | 5.5 | 12.7 | 12.8 | 0.6 | 2.5 | 24.0 | 13.2 | 20.7 | 49.4 |
| LVS | 1.3 | 9.7 | 23.6 | 2.0 | 6.5 | 12.1 | 5.0 | 9.4 | 16.0 | 0.7 | 3.8 | 8.3 | 1.0 | 4.2 | 8.8 | 6.2 | 16.4 | 40.5 | 5.4 | 12.7 | 12.2 | 0.5 | 3.9 | 25.3 | 12.5 | 22.4 | 50.6 |
| CAVP† | 3.1 | 15.2 | 38.7 | 3.8 | 7.0 | 13.2 | 5.5 | 10.2 | 18.4 | 0.9 | 4.7 | 8.4 | 1.2 | 4.9 | 9.8 | 8.4 | 21.2 | 44.5 | 6.6 | 16.5 | 36.3 | 2.9 | 7.5 | 20.4 | 10.5 | 23.0 | 39.5 |
| AVSegformer† | 2.7 | 15.5 | 39.5 | 3.6 | 7.1 | 12.8 | 5.3 | 9.8 | 17.2 | 0.9 | 4.5 | 7.9 | 1.1 | 4.6 | 9.4 | 8.0 | 20.3 | 29.9 | 8.7 | 17.8 | 32.6 | 2.5 | 7.3 | 20.2 | 11.2 | 23.7 | 40.9 |
| ImageBind | 1.4 | 5.8 | 14.5 | 1.2 | 5.7 | 11.3 | 3.4 | 7.2 | 12.8 | 0.3 | 3.1 | 6.6 | 0.4 | 3.8 | 7.5 | 5.8 | 14.5 | 27.2 | 4.2 | 16.3 | 24.0 | 0.5 | 2.7 | 15.3 | 6.5 | 12.4 | 25.1 |
| LanguageBind | 1.1 | 4.6 | 12.7 | 1.1 | 5.4 | 10.7 | 3.2 | 6.9 | 10.6 | 0.2 | 3.0 | 6.5 | 0.4 | 3.7 | 7.6 | 5.6 | 13.2 | 24.9 | 4.1 | 13.8 | 24.8 | 0.4 | 2.1 | 13.0 | 6.2 | 10.8 | 22.5 |
| EchoPin | 11.2 | 50.9 | 76.0 | 16.6 | 21.9 | 26.4 | 11.8 | 54.0 | 77.2 | 10.0 | 16.9 | 17.9 | 4.0 | 20.2 | 40.8 | **26.3** | **39.0** | **53.7** | 13.5 | 17.6 | 21.9 | 4.5 | 24.1 | 47.1 | 37.5 | 53.8 | 64.2 |
| Human | **59.3** | **79.3** | **90.7** | **40.0** | **57.9** | **67.9** | **54.3** | **65.7** | **80.0** | **9.3** | **49.4** | **76.8** | **12.3** | **45.1** | **80.3** | | - | | | - | | **25.7** | **36.4** | **38.6** | **60.9** | **82.8** | **89.1** |

Table S4: **Comparison of AI models in our benchmark across six conditions and three object sizes.** Bold and underlined values indicate the best and second-best performances, respectively. Since CAVP and AVSegformer are trained on MS-COCO2017, we evaluate them on a subset of AudioCOCO and denote their results with the symbol †.

