# OpenReview forum: "Seeing Sound, Hearing Sight: Uncovering Modality Bias and Conflict of AI models in Sound Localization"
_NeurIPS.cc/2025/Conference — NeurIPS 2025 spotlight_

### Official Review · Reviewer_fYXa · 2025-06-23

**Clarity:** 3
**Significance:** 4
**Originality:** 4
**Rating:** 5
**Confidence:** 4

**Summary:**

The authors created a visual-sound source localization dataset (AudioCOCO) and benchmarked ten AI models and human. The "Random" model serves as the low bound and performance from human psychophysics experiment serves as the high bound. This AudioCOCO dataset includes six conditions. Among those conditions, especially the conflicting visual cue (ConflictVcue), has almost never been tested by others.

**Questions:**

1. In the Abstract (L24) and Introduction (L91), the authors mentioned "physical structure of sensory systems". Does this sentence indicate the spectral cues created by human or animal's pinna? Unlike the AI model from Francl and McDermott which has two human-like ears and HRTF, I did not find any similarity between this paper's model and human auditory systems.

2. L72-73: Stereo audio, widely studied in neuroscience for its role in spatial hearing through inter-channel differences, remains underexplored in SSL benchmarks in AI. Do you mean all the SSL beachmarks in AI only use mono audio?

3. L84-86: Neuroscience research [3] shows that humans exhibit greater precision in horizontal (left-right) sound localization than vertical (up-down). Could you point out which sentence or figure in your reference support this? You referred a pure modeling paper without any human psychophysics experiments. There are lots of papers in SSL to support this, for example, JC Makous, JC Middlebrooks - The journal of the Acoustical Society of America, 1990.

4. In the your Figure 8, the performance of IS3 model is so bad even under the audio-visual congruent condition. I don't understand this since IS3 was published on 2023 and achieved SOTA during that time.

**Ethical Concerns:**

["NO or VERY MINOR ethics concerns only"]

**Final Justification:**

The authors have clarified my questions on the neuroscience part of this work.

**Limitations:**

Yes

**Quality:**

3

**Strengths And Weaknesses:**

I really like the idea of comparing the task performance and bias of AI models against humans or non-humans. Figures 4 and 5 and Table 1 clearly shown how the AI models fail under challenging conditions. This finding will benefit people from both AI and neuroscience field.

With that being said, the benefit I take from this paper is very limited. This model behaves very differently from human. In the L20-22 of Abstract, quote "Remarkably, consistent with neuroscience findings, our model exhibits a human-like horizontal localization bias—favoring left-right over up-down precision." The sentence refers to Figure 6 where the human has 52.1% at horizontal plane and 31.5% at vertical plane, so the accuracy dropped around 40%. In contrast, the SOTA AI model, IS3-FT-Stereo, has less than 10% of drop (the authors do not provide values). I do not think this single, very small, and likely insignificant change could be highlighted as finding the correspondance between AI and humans.

More importantly, the same Figure 3 also shows that the accuracy of both AI models on the vertical plane is much better than human. This is not suprsing to me at all: human relys on monaural spectral cues but AI models can still use ITD/ILD cues for localization sound on the vertical plane.

If the authors really want to claim they found human-like sound localization behaviors, they should check their third referred paper (A Francl, JH McDermott - Nature human behaviour, 2022) and show us compariable evidence as shown in that paper. Since the primary area of this paper is neuroscience instead of datasets or machine/deep learning, the strong connection with neuroscience is necessary.

In the first sentence of Introduction (L27-28), quote "Sound source localization (SSL) is the task of identifying the spatial origin of a sound within a visual scene." This is simply wrong. By this definitation, SSL will fail if sound occur in the darkness or from the human or animal's back. The third sentence (L29-30), quote "SSL requires integrating information across auditory and visual modalities". Multisensory integrations is beneficial but not required for SSL. I suggest the authors to revise many sentences like this which imply that SSL needs visual inputs.

---

> ### Author Rebuttal · Authors · 2025-07-31
>
> Thank you for the detailed review and thoughtful feedback on our work. Below, we address the concerns and respond to the reviewer’s questions point by point.
>
> **Weakness1: few benefits; model diverges from humans, direction gap is trivial.**
>
> Yes, the reviewer is correct in noting that the asymmetry bias between vertical and horizontal localization precision in IS3-FT-Stereo is relatively small—approximately a 4.6% difference (**Table R1**). To address the reviewer’s question more rigorously, we performed a two-tailed t-test comparing vertical and horizontal localization for IS3-FT-Stereo. The resulting p-value < 0.001, indicating a statistically significant difference between the two conditions.
>
> Furthermore, we conducted an ANOVA to compare vertical and horizontal localization biases across both human participants and IS3-FT-Stereo(p-value < 0.001). This analysis reveals that the asymmetry bias is significantly more pronounced in humans than in IS3-FT-Stereo.
>
> We agree with the reviewers that IS3-FT-Stereo represents a relatively simple extension of the IS3 model and is only loosely inspired by neuroscience at the high level. It lacks key features of the human auditory system, such as the use of monaural spectral cues, the complex structure of the pinna, and the anatomical organization of the cochlea for processing auditory signals. Despite these limitations, it is nonetheless noteworthy that IS3-FT-Stereo exhibits some degree of behavioral alignment with human localization performance—even if the effect is weaker than in more biologically plausible models, such as [3] in the paper.
> We will include these analyses and clarifications on neuroscience plausibility in the final version. We will also tone down the claim on asymmetry in audio perception in IS3-FT-Stereo.
>
> **Weakness2: AI uses ITD/ILD**
>
> Another excellent point! We agree with the reviewer that IS3-FT-Stereo performs considerably better than humans in the vertical plane, likely due to the model's advantage in utilizing ITD/ILD cues. Following the reviewer’s suggestion, we integrated human-like pinna and HRTF into IS3-FT-Stereo. As a result, the model's advantage in vertical localization is reduced! Please see **Weakness3** for details.
>
> **Weakness3: Reproducing Francl's evidence and bolstering neuroscience links.**
>
> As suggested by the reviewer and following the implementation in Francl(2022), we converted each stereo track in AudioCOCO into a 64-band time-frequency cochleagram using the PyCochleagram[3]. These cochleagrams were then used to fine-tune the IS3-FT-Stereo model, resulting in a new variant we call IS3-FT-Cochleagram. Results are presented in **Table R5**. For ease of comparison, we also include results from IS3-FT-Stereo and human participants(copied from the main paper).
>
> From **Table R1**, we observe that IS3-FT-Cochleagram exhibits stronger behavioral alignment with humans than IS3-FT-Stereo. In particular, the asymmetry between vertical and horizontal localization biases becomes more pronounced, and vertical biases—reliant on monaural spectral cues from the pinna—are more prominent. We will include these additional results and discussions in the final version.
>
> To further stress-test our core claim—that the physical structure of the sensory system shapes the multi-modal representations—we conducted an additional manipulation. We altered the input “diet” of IS3-FT-Stereo by rotating the human-like ears by 90 degrees. Specifically, we rotated the 3D coordinates in the Unity scene such that the physical left-right axis aligned with the image’s vertical axis, effectively simulating vertically arranged ears. We then re-rendered all stereo audio from this configuration. This variant is referred to as IS3-FT-Stereo-Rotated.
>
> Interestingly, in the IS3-FT-Stereo-Rotated condition, the asymmetry reverses, albeit with a relatively small effect size. This finding reinforces our central argument: the observed asymmetry in localization precision arises from the model's binaural receiver layout rather than any image-level artifact, supporting the idea that the physical structure of the sensory system directly shapes the multi-modal representations. Again, we acknowledge the absence of a pinna-like structure in IS3-FT-Stereo. To address this, we plan to modify the PyCochleagram to simulate a “90-degree rotated ear” configuration, and we will include the updated results in the final version of the paper.
>
> **Table R5. Vertical and horizontal localization bias for models and humans**
> |Method|Vertical|Horizontal|
> |-|:-:|:-:|
> |IS3-FT-Stereo|40.9|45.5|
> |IS3-FT-Stereo-Rotated|45.0|41.1|
> |IS3-FT-Cochleagram|34.7|51.8|
> |Human|31.5|52.1|
>
> **Weakness4:SSL definition**
>
> Thank you—this is another excellent point. We will revise our broad definition of SSL to identify the spatial origin of a sound in a 3D environment. We agree with the reviewer that vision is not a necessary condition for SSL.
> In this work, our focus is on studying multi-modal conflicts and biases, and therefore we specifically examine scenarios in which visual input is present. However, we acknowledge that conditions where auditory signals originate outside the visual field represent a compelling and important direction for future exploration. We will highlight this point in the future work section and revise all related statements about SSL throughout the manuscript for clarity and accuracy.
>
> **Question5: Clarify for model about the evident physiological analogue.**
> See the response to **Weakness3**.
>
> **Question6: SSL benchmarks**
>
> According to the strict definition of SSL, audio-only localization has been the focus of many AI datasets—such as DCASE[a] and LOCATA[b]—and corresponding models[c-e] that use stereo audio for computational modeling and benchmarking.
> However, since our focus lies in multimodal representation learning, we primarily rely on multimodal benchmark datasets, including VGG-SS[f], MUSIC[g], and Flickr-SoundNet[h]. These datasets predominantly use monaural audio, where visual input often compensates for the lack of spatial cues in the auditory stream. Notably, most popular multimodal models—such as AVSegformer, CAVP, IS3, and others—also rely on mono audio as the primary input modality. This trend reinforces the relevance of our evaluation design and underscores the untapped potential of stereo audio for capturing richer spatial information in complex multimodal settings.
> Motivated by our interest in modeling audio-visual conflicts, we deliberately explored both stereo and mono configurations for the audio input and conducted comparative analyses against leading mono-audio visual localization models. These comparisons further highlight the importance and benefit of incorporating stereo audio when addressing more challenging and ecologically valid scenarios in multi-modal learning.
>
> **Questions7:More Ref**
>
> We refer to Figures 4a and 4b in [3], which demonstrate that when listeners are forced to hear through altered ears, elevation localization degrades, while azimuthal localization remains largely unaffected. This finding suggests that azimuthal localization is robust due to its reliance on ITD and ILD, whereas elevation localization depends heavily on individual-specific spectral cues shaped by the pinna. Disrupting these cues leads to significantly reduced performance in estimating sound elevation.
> Thanks for the reference! The amazing study by Makous et al. provides a more direct and stronger evidence of localization bias through human psychophysics experiments. They showed that localization accuracy is better in the horizontal than vertical dimension for stimuli near the frontal midline, but that the reverse is true for stimuli located further in the periphery. Since our work focuses on audiovisual story perception within the visual field, which primarily spans the frontal midline, the localization bias toward better horizontal accuracy still holds. We will cite this work and clarify this point in the final version of the paper.
>
> **Question8: IS3 performance**
>
> We used the original implementation of IS3[i]. The primary reason for the poor performance of IS3 lies in the fact that many existing multi-modal methods inherit biased audio-visual representation alignments from their training datasets. As discussed in both the Introduction and Results sections, these models are heavily influenced by dataset bias, particularly the tendency for training data to center large, salient objects. Consequently, even state-of-the-art models struggle to accurately detect small or peripheral objects in a scene.
> This limitation is further supported by prior work[j-k], which highlights that the spatial location of the target plays a critical role in localization performance. These findings align with our observations and underscore the importance of addressing spatial bias when developing more robust and generalizable multimodal models.
>
> Ref:
>
> [a] Archontis et al.(2022) — STARSS22: A dataset of spatial recordings of real scenes with spatiotemporal annotations of sound events.
>
> [b] Löllmann et al.(2018) — The LOCATA Challenge Data Corpus for Acoustic Source Localization and Tracking
>
> [c] Hogeon Yu(2025) — A Two-Step Learning Framework for Enhancing Sound Event Localization and Detection
>
> [d] Kazuki et al.(2025) — Stereo sound event localization and detection with onscreen/offscreen classification
>
> [e] da et al.(2024) — SELD-Mamba: Selective State-Space Model for Sound Event Localization and Detection with Source Distance Estimation
>
> [f] Honglie et al.(2021) — Localizing Visual Sounds the Hard Way
>
> [g] Guangyao et al.(2022) — Learning to Answer Questions in Dynamic Audio-Visual Scenarios
>
> [h] Relja et al.(2017) — Look, listen and learn
>
> [i] Senocak et al.(2025) Toward Interactive Sound Source Localization: Better Align Sight and Sound
>
> [j] xizhou et al.(2018) Deformable ConvNets v2: More Deformable, Better Results
>
> [k] Antonio et al.(2011) Unbiased Look at Dataset Bias

---

> > ### Comment · Reviewer_fYXa · 2025-08-01
> > **Short question on new "IS3-FT-Cochleagram"**
> >
> > I appreciate the authors for their rebuttal letter which has addressed most of my questions on the AI part of this paper.
> >
> > The new "IS3-FT-Cochleagram" model shows very similar performance bias (horizontal vs vertical) as human listeners. I found this is very interesting and also surprising. So the only change is adding the PyCochleagram after the two raw audio inputs? Do you need to change the 1D-CNN (shown in Figure 3) to 2D-CNN?
> >
> > Regarding your rebuttal on **"Questions7:More Ref"**, the Figures 4a and 4b in your cited paper (Francl and McDermott, 2022) copied these two figrues (mentioned in their Figure legends) from the original experimental paper (Hofman, JGA Van Riswick, AJ Van Opstal - Nature neuroscience, 1998). Again, there is **NO** experimental results in Francl and McDermott, 2022.

---

> > > ### Author Response · Authors · 2025-08-01
> > >
> > > Thank you for your feedback and suggestions. Below, we address each of the reviewer’s additional questions.
> > >
> > > **Question1: The new "IS3-FT-Cochleagram" model shows very similar performance bias (horizontal vs vertical) as human listeners. I found this is very interesting and also surprising. So the only change is adding the PyCochleagram after the two raw audio inputs? Do you need to change the 1D-CNN (shown in Figure 3) to 2D-CNN?**
> > >
> > > The PyCochleagram library is used to convert stereo time-domain audio into a 2D cochleagram representation of size 133 × 160,000 × 2. Here, 133 refers to the number of frequency channels, derived from 64 base filters with a sampling factor of 2 (yielding 128 filters for overcomplete coverage), plus 5 additional filters to handle frequency boundaries and improve robustness to edge effects. The dimension 160,000 corresponds to the temporal resolution after processing a 10-second audio clip sampled at 16 kHz, and the final dimension of 2 captures the stereo channels. This 2D representation inherently preserves stereo information and is compatible with 2D convolutional neural networks, where 2D kernels are applied across the stereo channel axis.
> > >
> > > In contrast, our IS3-FT-Stereo approach retains stereo cues by using a 1D CNN-based audio branch that extracts spatial features jointly from both stereo channels at each time step. These features are then transformed using a short-time Fourier transform (SFFT) into a 2D spectral representation, making them compatible with 2D CNN encoders.
> > >
> > > We will clarify these points in the final version of the paper.
> > >
> > > **Question2: Regarding your rebuttal on "Questions7:More Ref", the Figures 4a and 4b in your cited paper (Francl and McDermott, 2022) copied these two figrues (mentioned in their Figure legends) from the original experimental paper (Hofman, JGA Van Riswick, AJ Van Opstal - Nature neuroscience, 1998). Again, there is NO experimental results in Francl and McDermott, 2022.**
> > >
> > > Thank you! We will cite the original experimental paper instead in the final version!

---

> > > > ### Comment · Reviewer_fYXa · 2025-08-01
> > > > **Cochleagram vs HRTF for vertical sound localization**
> > > >
> > > > Thanks for your clarification.
> > > >
> > > > I don't understand why add the cochleagram can introduce the human-like horizontal-vertical asymmetry shown in your Figure 6. This asymmetry was caused by the different localizaton cues used: ITD/ILD cues for horizontal (spectral cues also contribute), and spectral cues for vertical. Spectral cues were generated by the head-related transfer function (HRTF) due to the pinna filtering and head/torso shadowing. That is why changed the pinna can disrupt the vertical but not horizontal localization (the Van Opstal paper mentioned to you earlier). I don't think your audio synthesis include the spectral cues (you added 0.17m interaural distance for ITD).
> > > >
> > > > The major contribution of cochleagram is the upper limit of phase locking. A lower upper limit will reduce the horizontal localization performance but not vertical (Figure 3D in Saddler, JH McDermott - Nature Communications, 2024). Do you think it make sense that adding this limit can increase (instead of decrease) your model's horizontal localization accuracy (from 45.5% ot 51.8%)?

---

> ### Author Response · Authors · 2025-08-05
>
> Thank you for your feedback and suggestions. Below, we address each of the reviewer’s additional questions.
>
> **Question1: I don't understand why add the cochleagram can introduce the human-like horizontal-vertical asymmetry shown in your Figure 6. This asymmetry was caused by the different localizaton cues used: ITD/ILD cues for horizontal (spectral cues also contribute), and spectral cues for vertical. Spectral cues were generated by the head-related transfer function (HRTF) due to the pinna filtering and head/torso shadowing. That is why changed the pinna can disrupt the vertical but not horizontal localization (the Van Opstal paper mentioned to you earlier). I don't think your audio synthesis include the spectral cues (you added 0.17m interaural distance for ITD).**
>
> The reviewer is correct that the synthesized audio from Unity does not include spectral cues. Instead, it only provides ITD and ILD based on a 0.17 m interaural distance. Following the reviewer’s helpful suggestion, during the rebuttal we post-processed the stereo waveforms from Unity by converting them into cochleagram representations using the PyCochleagram library. This approach preserves the original ITD/ILD cues while adding spectral cues generated by the HRTF. These additional cues improve the IS3-FT-Cochleagram model’s A-Acc compared to IS3-FT-Stereo, as shown in the following Table R6.
>
> **Table R6. Model performance in terms of A-Acc for Audio Only condition within object size2**
> ||IS3-FT-Stereo|IS3-FT-Cochleagram-320Hz|IS3-FT-Cochleagram-3KHz|Human|
> |-|-|-|-|-|
> |A-Acc(%)|12.9|19.5|24.4|29.6|

---

> > ### Comment · Reviewer_fYXa · 2025-08-05
> > **PyCochleagram does not contain spectral cues**
> >
> > Thanks.
> >
> > In the 1st rebuttal, I asked "So the only change is adding the PyCochleagram after the two raw audio inputs?".
> > In the 2nd rebuttal, I asked about the HRTF for vertical localization. The rationale behind these two questions is that PyCochleagram has nothing to do with HRTF, it just converts audio waveforms into spectrogram using many ERB (equivalent rectangular bandwidth) banks borrowed from human cochlear. PyCochleagram (or Matlab version) was not developed for sound localization (I think JH McDermott developed this for his sound texture work in McDermott and Simoncelli, Neuron, 2011).
> >
> > What missing between raw audios and PyCochleagram is a HRTF that can simulate the pinna/head/torso (please refer to the Methods: Virtual acoustic simulateor: HRIRs in Francl and McDermott, 2022 for details). Therefore, I disagree "*This approach preserves the original ITD/ILD cues while adding spectral cues generated by the HRTF*." Something except spectral cues was added because A-Acc(%) changed. I don't know why.

---

> > > ### Author Response · Authors · 2025-08-06
> > >
> > > Thank you for pointing this out.
> > >
> > > We realized that we made a mistake in our earlier response by mistakenly assuming that HRTF‑based rendering was already incorporated into the PyCochleagram library. Following the reviewer’s guidance, we revisited the virtual acoustic simulator described in Francl & McDermott (2022) and traced it back to the underlying dataset of head‑related transfer function (HRTF) measurements from a KEMAR dummy head microphone (Gardner, 1995), derived from measured impulse responses.
> > >
> > > To properly incorporate HRTFs into our model, we now convolve the stereo audio recorded from Unity with the measured HRTFs in the Gardner dataset to simulate the directional filtering effects of the head, torso, and ears. The resulting HRTF‑processed waveforms are then passed into the PyCochleagram library, after which the model is fine‑tuned. This procedure ensures that both spatial filtering from HRTFs and cochleagram‑based auditory processing for ITD and ILD cues are included. We are currently running the updated experiments and will share the new A‑Acc and localization precision results as soon as they are available.
> > >
> > > Regarding the reviewer’s question about why A‑Acc increases even without HRTF, a possible explanation is that in IS3‑FT‑Stereo we use a standard Fourier transform with equal‑width frequency bins, whereas IS3‑FT‑Cochleagram employs ERB filters with non‑uniform bandwidths modeled after the human cochlea. This representation preserves phase coherence in low‑frequency channels for more reliable ITD extraction, aligns with reduced phase sensitivity at high frequencies to better capture ILDs, minimizes phase distortion in perceptually important ranges, suppresses irrelevant spectral detail, and mirrors the cochlea’s adaptive resolution. Together, these properties act as a localization‑specific inductive bias and can improve performance even without explicit HRTF processing.
> > >
> > > This has been a very useful and stimulating conversation with the reviewer! We sincerely appreciate the reviewer’s guidance, which has helped us gain a deeper understanding of auditory spatial processing in the brain.

---

> > > > ### Comment · Reviewer_fYXa · 2025-08-07
> > > > **Score updated and extra comments**
> > > >
> > > > I have updated my score from Reject to Borderline Accept. I also updated the scores of Quality, Clarity, and Significance all from 2 to 3.
> > > >
> > > > Comparing sound (or multimodal) localization behaviors in humans/animals with AI models will benefit both fields. To my best knowledge, there are only two papers about single modality (Francl/Saddler and McDermott, 2022/24), and no paper yet on multimodality localization. As long as the experiments (both psychophysics and model training) are executed in the right way, regardless of the results, I will recommed Accept or Strong Accept. However, I still have two major concerns:
> > > > 1. Huamn psychophysics. I did not ask questions on this part because it is infeasible to address this during the short rebuttal period. As we discussed earlier, I don't understand why an AI model without HRTF can localize vertical sound. **The same question holds true for the human subjects**. To localize vertical sound, aka to receive spectral cues, it requires either 1) in the free-field condition with multiple real speakers (Fig. 3c in Saddler and McDermott, 2024) or 2) measure the HRTF of each subject and present sounds through a VAS (virtual acoustic space). In this paper, the subjects just wearing headphones that only provide ITD/ILD cues but not spectral cues. The authors should explain this or add control studies in future.
> > > > 2. Results interpretations. The authors added the Cochleagram that is supposed to change only the horizontal localization. However, they observed changes in the vertical localization in the models. The authors then explained this in an entire paragraph "*Inside the cochlea, multiple coding mechanisms work together....Together, these mechanisms enable the brain to accurately determine where sounds come from in space*". I don't understand what they are talking about. **Those sentences are awkward without referring to a single paper**. I wish to read convincing interpretations in future.
> > > >
> > > > All in all, this is an important work that I believe will benefit both cognitive/systems neuroscience and AI.
> > > > Great job!

---

> > > > > ### Author Response · Authors · 2025-08-08
> > > > >
> > > > > **Question1: Huamn psychophysics. I did not ask questions on this part because it is infeasible to address this during the short rebuttal period. As we discussed earlier, I don't understand why an AI model without HRTF can localize vertical sound. The same question holds true for the human subjects. To localize vertical sound, aka to receive spectral cues, it requires either 1) in the free-field condition with multiple real speakers (Fig. 3c in Saddler and McDermott, 2024) or 2) measure the HRTF of each subject and present sounds through a VAS (virtual acoustic space). In this paper, the subjects just wearing headphones that only provide ITD/ILD cues but not spectral cues. The authors should explain this or add control studies in future.**
> > > > >
> > > > > We sincerely apologize for the earlier oversight. Upon re-examining our Unity-based 3D audio rendering framework, we realized that the Unity Audio Spatializer SDK we used already incorporates the default KEMAR HRTF. This means our stimuli included not only interaural time and level differences (ITD/ILD), but also pinna-related spectral cues.
> > > > >
> > > > > In fact, the HRTF implemented in Unity is based on the default KEMAR dataset. For the reviewer’s reference, a relevant excerpt from Unity’s documentation states:
> > > > >
> > > > > **The HRTF filtering is based on a modified version of the KEMAR data set. For more information on the KEMAR data set, see the MIT Media Lab’s documentation and measurement files.**
> > > > >
> > > > > We will clarify this point explicitly in the revised manuscript:
> > > > >
> > > > > **We used Unity to render 3D audio based on 3D visual information and applied the default HRTF filtering provided by Unity, which is adapted from the KEMAR dataset developed at the MIT Media Lab. The resulting binaural audio was then transformed into a cochleagram using multiple equivalent rectangular bandwidth (ERB) filters, following the human cochlear model implemented in the PyCochleagram library.**
> > > > >
> > > > > For reproducibility, we will release all code, data, and trained models upon publication—including the MATLAB scripts used for the human psychophysics experiments and their integration with the Unity simulator.
> > > > >
> > > > > We also greatly appreciate the reviewer’s neuroscience perspective on human psychophysics methodology. As suggested, in future experiments involving auditory perception, we plan to measure individualized HRTFs for each participant and present the sounds using a virtual auditory space (VAS) setup.
> > > > >
> > > > > **Question2: Results interpretations. The authors added the Cochleagram that is supposed to change only the horizontal localization. However, they observed changes in the vertical localization in the models. The authors then explained this in an entire paragraph "Inside the cochlea, multiple coding mechanisms work together....Together, these mechanisms enable the brain to accurately determine where sounds come from in space". I don't understand what they are talking about. Those sentences are awkward without referring to a single paper. I wish to read convincing interpretations in future.**
> > > > >
> > > > > With our corrections explained above where the audio spatial localizer SDK in Unity involved spectral cues, that explains why the model can vertically localize the objects. We will include the relevant neuroscience references in the revised manuscript to provide proper context and support for all the claims in the paper. We sincerely thank the reviewer once again for the stimulating and insightful discussion throughout the rebuttal process! It has been an incredibly enriching experience, and we’ve learned a great deal!

---

> > > > > > ### Comment · Reviewer_fYXa · 2025-08-08
> > > > > > **Score updated again**
> > > > > >
> > > > > > I just read the manual of unity3d to fact check the authors claim. Yes, they are correct.
> > > > > >
> > > > > > This explained the vertical bias observed in both humans and AI models. Therefore, I think the experiments are executed in the right way. As promised in my last comment, I recommed Accept (I also updated the score of Significance and Originality to excellent).

---

> > > > > > > ### Author Response · Authors · 2025-08-09
> > > > > > > **Thanks for your revisions and suggestions.**
> > > > > > >
> > > > > > > Dear Reviewer,
> > > > > > >
> > > > > > > Thank you for your recognition of this work.
> > > > > > >
> > > > > > > Best regards
> > > > > > >
> > > > > > > Authors

---

> ### Author Response · Authors · 2025-08-05
>
> **Question2: The major contribution of cochleagram is the upper limit of phase locking. A lower upper limit will reduce the horizontal localization performance but not vertical (Figure 3D in Saddler, JH McDermott - Nature Communications, 2024). Do you think it make sense that adding this limit can increase (instead of decrease) your model's horizontal localization accuracy (from 45.5% ot 51.8%)?**
>
> We would like to clarify that Table R5 and Figure 6 do not report Audio Localization Accuracy (A‑Acc). Instead, they present the within‑6° proportion—the percentage of predicted points falling within ±6° of visual angle along a given axis from the ground‑truth bounding box center. This metric measures spatial precision along a single axis, whereas A‑Acc measures whether the predicted location lies anywhere within the full segmentation mask of the sounding object. The two metrics are distinct and need not correspond directly.
>
> For example, in the human evaluation with 717 trials, 225 predictions fell within ±6° vertically and 374 horizontally. Of these, 212 predictions also lay within the segmentation mask. Thus, the within‑6° proportion is 31.5% (225/717) vertically and 52.1% (374/717) horizontally, while the corresponding A‑Acc is 29.6% (212/717).
>
> The reviewer correctly points out that the outer ear (pinna) shapes incoming sounds by filtering frequencies based on their elevation, creating spectral cues essential for vertical localization. Inside the cochlea, multiple coding mechanisms work together: phase locking allows auditory nerve fibers to synchronize with low-frequency sound waves, supporting horizontal localization through timing differences; place coding maps frequencies along the cochlea’s basilar membrane, representing both pitch and the pinna’s spectral patterns for vertical localization. Additionally, envelope coding tracks slower amplitude fluctuations, aiding in processing speech rhythms and sound separation, while rate coding conveys the average firing rate of neurons, signaling sound intensity and supporting localization. Together, these mechanisms enable the brain to accurately determine where sounds come from in space.
>
> Following the reviewer’s suggestion, we examined the phase‑locking mechanism in the IS3‑FT‑Cochleagram model, using configurations from [a, b, c]. We compared cochleagram inputs with phase‑locking limits of 320 Hz and 3 kHz. As shown in Table R6, extending the upper limit from 320 Hz to 3 kHz yields a clear performance gain: A‑Acc increases from 19.5% to 24.4%. This suggests that biologically inspired phase locking enhances temporal resolution in the cochleagram, thereby improving localization accuracy.
>
> Table R7 reports the within‑6° proportions for humans, IS3‑FT‑Cochleagram‑320Hz, and IS3‑FT‑Cochleagram‑3kHz. Lowering the upper limit typically reduces horizontal localization precision (from 51.8 % to 47.7 %), while vertical precision remains largely unaffected (34.7 % vs. 36.2 %). This aligns with auditory biology: reducing phase‑locking bandwidth degrades timing cues critical for horizontal localization. The A‑Acc improvement from 320 Hz to 3 kHz is thus largely driven by gains in horizontal precision.
>
> Notably, IS3‑FT‑Stereo—without any cochleagram mechanism—performs the worst. In contrast, IS3‑FT‑Cochleagram‑3 kHz achieves vertical and horizontal localization precision that closely matches human performance, including the larger gap between horizontal and vertical precision observed in humans. This underscores the importance of a human‑like cochleagram representation. All new results and discussions will be incorporated into the final version.
>
> **Table R7. Vertical and horizontal localization bias for models and humans for Audio Only condition within object size2**
> |Method|Vertical|Horizontal|
> |-|-|-|
> |IS3-FT-Stereo|40.9|45.5|
> |IS3-FT-Cochleagram-320Hz|36.2|49.0|
> |IS3-FT-Cochleagram-3KHz|34.7|51.8|
> |Human|31.5|52.1|
>
> References
>
> [a] Mrak et al.(2024) Models optimized for real-world tasks reveal the task-dependent necessity of precise temporal coding in hearing
>
> [b] Mrak et al.(2021) Deep neural network models reveal interplay of peripheral coding and stimulus statistics in pitch perception
>
> [c] Bruce et al.(2018) A phenomenological model of the synapse between the inner hair cell and auditory nerve: Implications of limited neurotransmitter release sites

---

### Official Review · Reviewer_AmaU · 2025-06-27

**Clarity:** 3
**Significance:** 3
**Originality:** 4
**Rating:** 5
**Confidence:** 3

**Summary:**

This paper investigates modality bias and conflict resolution in multimodal AI systems for sound source localization (SSL). Through a carefully designed psychophysics-inspired framework, the authors benchmark human and AI performance across six audio-visual conditions (e.g., congruent, conflicting, absent cues). They introduce *AudioCOCO*, a large-scale dataset synthesized via 3D simulation, containing stereo audio rendered from static images. Their experiments reveal that current AI systems are heavily biased toward visual cues and struggle under conflicting or missing modality conditions. By fine-tuning a state-of-the-art model on their new dataset with stereo input, they achieve significant performance gains and observe human-like spatial biases, such as better horizontal than vertical localization.

**Questions:**

Please refer to the 'weakness' part.

**Ethical Concerns:**

["NO or VERY MINOR ethics concerns only"]

**Final Justification:**

The authors' rebuttal addressed my main concerns. Overall, I think the stereo audio-involved sound source localization is a novel and interesting problem. The proposed large-scale dataset is well-designed. Therefore, I would like to increase my final rating.

**Limitations:**

Yes, the authors have clearly articulated limitations, including dataset realism and the restriction to static images. They also discuss potential directions to incorporate video and better audio realism in future work.

**Quality:**

3

**Strengths And Weaknesses:**

## Strengths

- The paper is clearly written with good motivation, explanations, and visualizations.
- Few existing works on sound source localization explore the modality bias, object bias, and the stereo audio. The proposed dataset with stereo audio simulation is novel, scalable, and would beneficial for the SSL community.
- Based on the proposed dataset, this paper also provided some insightful conclusions which may inspire future works.

## Weaknesses





- **Unclear evaluation metrics.** 1) The authors introduce the evaluation metrics of 'A-Acc' and 'V-Acc' at Lines 235-242. As shown in Table S1, only one metric is reported for some SSL cases. This raises questions on the application and suitable scenarios of the metrics. Could the authors provide more explanations on this? 2) Prior SSL works usually adopt the CIoU to evaluate the model's performance. Can this metric be used in this paper? If not, could the authors provide some discussion on this?
- **Finetuning and Inference time cost**. The proposed dataset is large-scale. I wonder about the time cost of fine-tuning the IS3 model and the inference.
- **Questions on experiments** 1) According to Table 1 (1), the CAVP has better performance than IS3, so can the CAVP also be adopted for Finetuning using the proposed dataset? This would make the comparison more comprehensive. 2) The authors choose to fine-tune the prior model IS3. Could the IS3 model be directly trained from scratch?
- **Minor issues on Figure presentation (do not affect my score).** 1) In Figure 1(a), the texts of 'left' and 'right' seem to be hard to correspond to the figure. 2) In Figure 3, there are no 'blue arrows' mentioned in the caption.

---

> ### Author Rebuttal · Authors · 2025-07-31
>
> Thank you for the detailed review and thoughtful feedback on our work. Below, we address the concerns point by point and respond to the reviewer’s questions individually.
>
> **AmaU.1 - Questions: Unclear evaluation metrics. 1) The authors introduce the evaluation metrics of 'A-Acc' and 'V-Acc' at Lines 235-242. As shown in Table S1, only one metric is reported for some SSL cases. This raises questions on the application and suitable scenarios of the metrics. Could the authors provide more explanations on this? 2) Prior SSL works usually adopt the CIoU to evaluate the model's performance. Can this metric be used in this paper? If not, could the authors provide some discussion on this?**
>
> Audio Accuracy (A-Acc) measures whether the model or human correctly localizes the true sound source regardless of semantic matching between audio and vision. According to the definition of sound source localization (SSL), the primary goal is to localize the location of the sound. However, in the Vision-only condition, there is no audio input. Hence, there is no definitive ground truth for A-Acc evaluation, since all possible locations in the image could be considered correct or incorrect. To address this, we introduce another metric called V-Acc, which measures alignment with visual semantics. This metric is only applicable in special conditions where there is no sound or when multiple visual instances match the sound. We will clarify this distinction in the final version.
> The reviewer also raises a good point regarding CIoU (Complete Intersection over Union), which is a common metric in computer vision. However, applying this metric in human psychophysics experiments is challenging because humans rarely draw bounding boxes around target objects during localization tasks. Moreover, it would be both difficult and unfair to require humans and models to draw bounding boxes on blank images under the audio-only condition, where no visual stimuli are present. We will clarify this point as well in the final version.
>
> **AmaU.2 - Questions: Finetuning and Inference time cost. The proposed dataset is large-scale. I wonder about the time cost of fine-tuning the IS3 model and the inference.**
>
> We contributed a large-scale AudioCOCO dataset containing roughly one million audio-image pairs by combining selected images and audios using a stringent set of filtering criteria (Section 2.1 and 2.2). As the reviewer correctly suspects, we did not use all possible combinations of image-audio pairs to fine-tune the IS3 models. Instead, we randomly sampled a small subset of 9,356 image-audio pairs for fine-tuning IS3.
> We used 4 A6000 GPUs to run the experiment over 10 epochs, with a total runtime of about 4 hours. Despite using only a small subset of the full training set, IS3-FT-Stereo has already demonstrated remarkable performance compared to other IS3 variants. This provides strong evidence emphasizing that data quality matters more than data quantity. We will clarify this point in the final version.
>
> **AmaU.3 - Questions: 1) According to Table 1 (1), the CAVP has better performance than IS3, so can the CAVP also be adopted for Finetuning using the proposed dataset? This would make the comparison more comprehensive. 2) The authors choose to fine-tune the prior model IS3. Could the IS3 model be directly trained from scratch?**
>
> We chose IS3 because it is a widely used baseline in the literature and achieves the best performance among all baselines under most conditions. Therefore, we use IS3 as the representative baseline for the main paper’s analyses. Additionally, we compare other baselines against humans and IS3-FT-Stereo, providing these results and analyses in the appendix due to space constraints.
> As the reviewer correctly noted, similar to IS3, CAVP and other audio-visual localization models can also be fine-tuned on our proposed AudioCOCO dataset. We have now fine-tuned CAVP on AudioCOCO and present a performance comparison in ** Table R4 **. Similar to the analysis on IS3, these results highlight consistent performance improvements attributable to both the high-quality data and the inclusion of stereo audio. We will include these findings in the final version of the paper.
> Despite these performance gains after fine-tuning on our dataset, it is important to note that AudioCOCO remains a synthetic dataset. While training models from scratch on AudioCOCO is certainly possible, we argue that leveraging pretrained weights—especially those obtained from large-scale, real-world datasets—offers substantial benefits. These pretrained weights encode rich semantic priors that enable models to generalize better.
> Loosely connected to the human learning process, our fine-tuning strategy reflects a similar approach: just as humans build upon years of real-world experience rather than learning each task in isolation, models can benefit from general pretraining followed by task-specific refinement. This fine-tuning helps to mitigate the domain gaps that often cause performance drops in AI models when compared with humans. Importantly, this does not imply that our analysis is confounded by domain gap issues. The comparison between IS3-FT-Mono and IS3-FT-Stereo shows gains of 4.4% for Size1, 4.4% for Size2, and 1.4% for Size3, highlighting the contribution of spatial cues derived from stereo audio.
> Overall, we advocate using foundation models pretrained on large-scale datasets that can be fine-tuned on domain-specific, high-quality datasets such as AudioCOCO to achieve optimal performance.
>
> Table 4. A-Acc on congruent conditions for CAVP and CAVP-FT-Stereo
> | Method            | Size 1 | Size 2 | Size 3 |
> |-------------------|:------:|:------:|:------:|
> | CAVP              | 5.1    | 15.2   | 38.7   |
> | CAVP-FT-Stereo    | 20.4 (+14.6 %) | 28.8 (+13.6 %) | 52.5 (+13.8 %) |
>
>
> **AmaU.4 - Questions: Minor issues on Figure presentation (do not affect my score). 1) In Figure 1(a), the texts of 'left' and 'right' seem to be hard to correspond to the figure. 2) In Figure 3, there are no 'blue arrows' mentioned in the caption.**
>
> Thank you! We will fix all the figure presentation issues accordingly.

---

> > ### Comment · Reviewer_AmaU · 2025-08-02
> >
> > Thanks to the authors for their rebuttal. The response has addressed my concerns regarding the evaluation metrics and the comparison with CAVP. Regarding Question 2, the authors mentioned that 9,356 image-audio pairs (out of 1,003,747 in the full dataset) were **randomly** selected for IS3 fine-tuning. Therefore, I am curious about the utility of the remaining data and how a fair comparison is ensured for future work, given the randomness in selecting the training subset. Moreover, could the authors provide more specific statistical analysis of the selected training set to better justify its representativeness? hope to hear more discussion from the authors. Thanks.

---

> > > ### Author Response · Authors · 2025-08-04
> > >
> > > Thank you for your feedback and suggestions. Below, we address your question.
> > >
> > > **Question: Regarding Question 2, the authors mentioned that 9,356 image-audio pairs (out of 1,003,747 in the full dataset) were randomly selected for IS3 fine-tuning. Therefore, I am curious about the utility of the remaining data and how a fair comparison is ensured for future work, given the randomness in selecting the training subset. Moreover, could the authors provide more specific statistical analysis of the selected training set to better justify its representativeness? hope to hear more discussion from the authors.**
> > >
> > >
> > > We would like to clarify that the AudioCOCO dataset consists of six conditions, but we only use the audio-visual congruent condition for fine-tuning. As described in Section 2.2 and Section S3, the congruent condition contains 184,227 image-audio pairs. We fine-tune the model on a random 5% subset of the congruent-condition training data, then evaluate it on an independent 5% test subset drawn from each of the six conditions. To reduce potential sampling bias at the dataset level, we repeated the process three times with different random seeds and report the averaged results with standard error of the mean (SEM). The models perform consistently well across all three runs with a small SEM.
> > >
> > >
> > > Compared to the original MSCOCO and VGG-Sound datasets, our AudioCOCO dataset has undergone a series of careful data filtering and data distribution balancing. We visualize the distributions of audio categories, spatial object locations, and image categories (Figures S1–S3) to confirm this. Our goal is to ensure that the dataset supports fair evaluation under modality bias and conflict conditions.
> > >
> > >
> > > We will release all data, code, and models for reproducibility purposes.

---

> > > > ### Comment · Reviewer_AmaU · 2025-08-04
> > > >
> > > > Thanks to the authors for their follow-up response. I have no major concerns remaining. Considering the value of the proposed dataset and the novelty of the explored problem, I would like to maintain my original rating. Also, I hope the authors can address other concerns from the knowledgeable reviewer fYXa.

---

### Official Review · Reviewer_BL6m · 2025-07-02

**Clarity:** 4
**Significance:** 3
**Originality:** 4
**Rating:** 4
**Confidence:** 3

**Summary:**

This paper presents a detailed and systematic analysis of modality bias and conflict in the task of sound source localization (SSL). The authors propose six possible audiovisual scenarios: Congruent, ConflictVcue, AbsVcue, Aonly, Vonly, and MultiInstLoc. They synthesize an audiovisual dataset by pairing semantically similar images and sounds using the MSCOCO image dataset and the VGGSound audio dataset. Depth information is inferred from the images using the DepthAnything model, and depth-aware stereo audio rendering is performed in Unity.

The authors fine-tune the state-of-the-art SSL model IS3 on their newly constructed AudioCOCO dataset, resulting in IS3-FT. In the experiments, both human psychophysics studies and quantitative evaluations are conducted. The results show that object size is positively correlated with sound localization accuracy; audiovisual scenarios involving modality conflict harm localization accuracy. Existing AI models struggle with sound source localization in audio-only settings, while achieving relatively good V-ACC in vision-only conditions—indicating a preference for visual information. Human performance surpasses that of AI models across all audiovisual scenarios, particularly under audio-only conditions. Both humans and AI models perform poorly on multi-instance SSL tasks. The fine-tuned model IS3-FT outperforms the original IS3 on AudioCOCO. Moreover, the stereo-audio-aware IS3-FT-stereo model outperforms its mono counterpart IS3-FT-mono, and displays human-like localization biases—favoring horizontal over vertical precision.Strengths And Weaknesses

**Questions:**

1. It is unclear whether the fine-tuning of the IS3 model was conducted solely on congruent data, or if modality conflict data was also used. If the latter, the introduction of ambiguous or conflicting audiovisual signals could potentially harm the model’s original performance. The paper should clarify whether modality conflict data was included during fine-tuning and analyze its effect on model robustness.

2. Does IS3-FT outperform IS3 on standard SSL test datasets such as VGG-SS and SoundNet-Flickr-Test? The paper currently lacks evaluations on these widely used benchmarks, making it difficult to assess the generalizability of the improvements reported on AudioCOCO.

3. The paper analyzes the impact of object size on SSL accuracy. However, based on the evaluation metric used for SSL accuracy—where a prediction is considered positive if the location of the maximum activation on the heatmap lies within the segmentation mask of the ground truth sounding object—it is important to note that larger objects naturally have a higher probability of being matched by chance. That is, even a randomly selected pixel from the image is more likely to fall within the object mask if the object is large (as supported by the authors' own experiment involving random sampling). Therefore, the higher SSL accuracy observed for larger objects may largely result from the limitations of the evaluation metric, rather than indicating a true model sensitivity to object size. While the conclusion that “Object size matters for humans and AI” may subjectively make sense, the current experimental design and analysis are insufficient to convincingly support this claim.

**Ethical Concerns:**

["NO or VERY MINOR ethics concerns only"]

**Limitations:**

Yes

**Quality:**

3

**Strengths And Weaknesses:**

a) Strengths

1.	The paper introduces a new and challenging benchmark, UniAV, for the SSL task. The authors conduct a detailed analysis of various possible audiovisual scenarios in SSL, with a particular focus on modality conflict—an interesting and meaningful case that addresses issues not previously considered in the literature and has practical relevance.

2.	A large-scale, high-quality audio-visual dataset is constructed, simulating realistic, spatialized stereo audio that obeys the physical laws of sound propagation. By fine-tuning existing models on this dataset, the authors achieve performance that surpasses the current state-of-the-art on the UniAV benchmark.

3.	The paper includes both psychophysics and quantitative experiments, analyzing the performance differences between existing AI models and humans in SSL tasks.

b) Weaknesses

1.	The synthesized dataset is based on static images rather than videos, which leads to a natural temporal inconsistency between the visual and audio information. This creates a gap between the benchmark and real-world sound source localization scenarios.

2.	The evaluation is primarily conducted on the authors' proposed dataset (AudioCOCO) and benchmark (UniAV), lacking assessments on commonly used benchmarks in the field. For example, the paper does not demonstrate whether IS3-FT outperforms IS3 on VGG-SS [1] and SoundNet-Flickr-Test [2]. Therefore, the claimed improvement over the state-of-the-art may result from overfitting to the AudioCOCO dataset.

3.	The paper lacks a review of related work.

[1] Honglie Chen, Weidi Xie, Triantafyllos Afouras, Arsha Nagrani, Andrea Vedaldi, and Andrew Zisserman. Localizing visual sounds the hard way. In IEEE Conference on Computer Vision and Pattern Recognition, 2021.

[2] Arda Senocak, Tae-Hyun Oh, Junsik Kim, Ming-Hsuan Yang, and In So Kweon. Learning to localize sound source in visual scenes. In IEEE Conference on Computer Vision and Pattern Recognition, 2018.

---

> ### Author Rebuttal · Authors · 2025-07-31
>
> Thank you for the detailed review and thoughtful feedback on our work. Below, we address the concerns point by point and respond to the reviewer’s questions individually.
>
> **BL6m.1 - Weakness: The synthesized dataset is based on static images rather than videos, which leads to a natural temporal inconsistency between the visual and audio information. This creates a gap between the benchmark and real-world sound source localization scenarios.**
>
> We agree with the reviewer that studying audio in videos is a natural direction, as both modalities inherently involve temporal information. We acknowledge this limitation in the Discussion section.
> Our decision to start with static images for studying sound source localization (SSL) was motivated by the relative scarcity of work focusing on modality conflict and bias. Static images provide better experimental control, allowing us to isolate key factors more precisely.
> In future work, we plan to extend our approach to video settings and warmly welcome the community—including the reviewer—to join us in this endeavor.
>
> **BL6m.2 - Weakness: The evaluation is primarily conducted on the authors' proposed dataset (AudioCOCO) and benchmark (UniAV), lacking assessments on commonly used benchmarks in the field. For example, the paper does not demonstrate whether IS3-FT outperforms IS3 on VGG-SS [1] and SoundNet-Flickr-Test [2]. Therefore, the claimed improvement over the state-of-the-art may result from overfitting to the AudioCOCO dataset.**
>
> We respectfully disagree with the reviewer’s suggestion that the performance improvement in IS3-FT-Stereo is due to overfitting on the AudioCOCO dataset. To clarify, we introduce three IS3 variants in the paper:
> First, IS3: The original model trained on the Flickr-SoundNet [53] and VGG-Sound [46] datasets.
> Second, IS3-FT-mono: The original model fine-tuned on our AudioCOCO dataset, using monaural audio as input.
> Third, IS3-FT-Stereo: The original model fine-tuned on our AudioCOCO dataset, using stereo audio channels as input.
> The performance gap between IS3-FT-mono and the original IS3 model is 3.3% for Size1 (small objects), 7.7% for Size2, and 9.3% for Size3, clearly demonstrating the positive effect of improved data quality. Furthermore, the comparison between IS3-FT-Mono and IS3-FT-Stereo shows gains of 4.4% for Size1, 4.4% for Size2, and 1.4% for Size3, highlighting the contribution of spatial cues derived from stereo audio, rather than the domain gap effect.
>  We will clarify this point explicitly in the final version.
> As suggested by the reviewer, we conducted an additional experiment by fine-tuning our IS3-FT-Stereo model using the VGG-SS and Flickr-SoundNet datasets, and report comparative results against IS3 and CAVP using the cIoU metric under the default evaluation protocol provided by [a]. From ** Table R3**, we observed that benefiting from the fine-grained spatial features provided by AudioCOCO, our IS3-FT-Stereo model achieves superior performance on these standard SSL benchmarks, outperforming the baselines. These results and discussions will be included in the final version of the paper.
>
> [a]. Arda Senocak et al. (2018) Learning to localize sound source in visual scenes.
> Table R3. Results in cIoU for models on VGG-SS and Flickr-SoundNet
> | Method        | VGG-SS | Flickr-SoundNet |
> |---------------|:------:|:---------------:|
> | IS3           | 42.96  | 84.40           |
> | CAVP          | 43.58  | 85.03           |
> | **IS3-FT-Stereo** | **43.79** | **85.65** |
>
> **BL6m.3 - Questions: It is unclear whether the fine-tuning of the IS3 model was conducted solely on congruent data, or if modality conflict data was also used. If the latter, the introduction of ambiguous or conflicting audiovisual signals could potentially harm the model’s original performance. The paper should clarify whether modality conflict data was included during fine-tuning and analyze its effect on model robustness.**
>
>
> We fine-tuned IS3 (IS3-FT) using only congruent condition data, with the goal of evaluating the model’s out-of-distribution (OOD) performance under five other novel and more challenging conditions—such as audio-visual conflicts.
>
> Training IS3-FT-Stereo on modality conflicts is not even valid. In such scenarios, there is often no definitive ground truth, as conflicting cues introduce inherent ambiguity. Localizing the car with a barking dog sound would confuse the models to learn correct visual semantic and audio semantic associations.
>
> Therefore, rather than training on these conditions, our paradigm in benchmarking model behavior when confronted with such real-world inconsistencies becomes more valuable.
> Conflict conditions are not pathological but common in natural environments (e.g., occlusions, off-screen sound sources, or mismatched audio and visual stimuli). Our long-term objective is to propose new methods capable of flexibly interpreting and resolving these complex scenes, building on the foundational contributions of this paper.
>
>
>
> **BL6m.4 - Questions: Does IS3-FT outperform IS3 on standard SSL test datasets such as VGG-SS and SoundNet-Flickr-Test? The paper currently lacks evaluations on these widely used benchmarks, making it difficult to assess the generalizability of the improvements reported on AudioCOCO.**
>
> See the response to ** BL6m.2 - Questions **.
>
> **BL6m.5 - Questions: The paper analyzes the impact of object size on SSL accuracy. However, based on the evaluation metric used for SSL accuracy—where a prediction is considered positive if the location of the maximum activation on the heatmap lies within the segmentation mask of the ground truth sounding object—it is important to note that larger objects naturally have a higher probability of being matched by chance. That is, even a randomly selected pixel from the image is more likely to fall within the object mask if the object is large (as supported by the authors' own experiment involving random sampling). Therefore, the higher SSL accuracy observed for larger objects may largely result from the limitations of the evaluation metric, rather than indicating a true model sensitivity to object size. While the conclusion that “Object size matters for humans and AI” may subjectively make sense, the current experimental design and analysis are insufficient to convincingly support this claim.**
>
> The reviewer correctly points out that raw A-Acc may be confounded by the area of the ground-truth mask—since, intuitively, smaller objects are more difficult to localize while larger ones are easier. To address this potential bias, we introduce a chance-corrected gain metric, which quantifies the improvement of a human or model over a random guess, normalized by the baseline accuracy of the random guess:
> $$
> \mathrm{Gain}(X) = \frac{\mathrm{Acc} _X - \mathrm{Acc} _{\text{rand}}}{\mathrm{Acc} _{\text{rand}}} \times 100 \%
> $$
>
> $\displaystyle \text{Acc}_{\text{rand}}$ represents the accuracy achieved when predictions are sampled uniformly at random across the scene.
>
> This normalization removes the influence of mask size and isolates the true localization capability of humans and models. As shown in ** Table R2 ** above, even after correcting for chance, humans consistently outperform models—especially for small object sizes. For example, in the smallest size category, humans achieve a gain of 17.9%, compared to only 1.7% for models.
> These results suggest that object size modulates sound source localization performance in a way that cannot be fully explained by ground-truth mask area alone, highlighting deeper perceptual and representational differences between human and model behavior.
>
>
> **BL6m.5 - Weakness:The paper lacks a review of related work.
> [1] Honglie Chen, Weidi Xie, Triantafyllos Afouras, Arsha Nagrani, Andrea Vedaldi, and Andrew Zisserman. Localizing visual sounds the hard way. In IEEE Conference on Computer Vision and Pattern Recognition, 2021.
> [2] Arda Senocak, Tae-Hyun Oh, Junsik Kim, Ming-Hsuan Yang, and In So Kweon. Learning to localize sound source in visual scenes. In IEEE Conference on Computer Vision and Pattern Recognition, 2018.**
>
> Thanks for recommending these two excellent works!
>
> [1] introduces the VGG-SS dataset, which extends VGG-Sound with fine-grained ground truth annotations to support audio-visual segmentation (AVS) tasks. It also proposes a method for mining hard samples and integrating them into contrastive learning pipelines to enhance model performance.
> [2] presents the Flickr-SoundNet dataset, aimed at enabling large-scale audio-visual learning and localization in unconstrained, real-world environments.
>
> While both datasets have significantly advanced benchmark development for sound source localization (SSL), they also face key limitations. Specifically, the original audio tracks are monaural, which limits the availability of spatial cues essential for accurate localization by AI models. In addition, neither dataset explicitly addresses location bias or category imbalance, both of which can hinder model generalization.
>
> Moreover, the hard sample mining strategy in [1] may be less effective—or even counterproductive—under multi-instance conflict scenarios, where overlapping or incongruent audiovisual cues are common. In such settings, contrastive learning objectives may be misled by ambiguous or conflicting signals.
>
> To overcome these challenges, we introduce a filtering metric to curate high-quality, spatially balanced samples in our AudioCOCO dataset. We also render stereo audio using Unity, allowing us to create more realistic, spatially informative scenes. This enables AudioCOCO to support a wider range of multi-modal reasoning tasks under both congruent and conflicting conditions.
>
> We will cite these important works and explicitly highlight the differences between them and ours in the final version of the paper.

---

> ### Author Response · Authors · 2025-08-09
>
> We sincerely appreciate your insightful feedback and constructive suggestions. Please let us know if you have any remaining concerns or additional suggestions.

---

> > ### Comment · Reviewer_BL6m · 2025-08-09
> >
> > Thank you for your detailed response and clarifications, which have addressed most of my concerns. Considering the strengths and weaknesses, I will keep my original score.

---

### Official Review · Reviewer_f4Cu · 2025-07-03

**Clarity:** 4
**Significance:** 3
**Originality:** 2
**Rating:** 4
**Confidence:** 4

**Summary:**

The paper focuses on audiovisual sound source localization. It is a study focussing on the interactions between the different modalities. Particularly interesting are cases where the two modalities have conflicting cues or if either is missing or noisy. They construct a new dataset from images in COCO and audio from VGGSound and smartly pair them. They use depth estimation models to construct stereo sounds. They compare results of AI models with a human study in similar conditions and derive interesting observations from them.

**Questions:**

I would gladly increase my rating if the authors can defend the choice of metric and address my concerns about its possible effects on the paper's key takeaways.

**Ethical Concerns:**

["NO or VERY MINOR ethics concerns only"]

**Final Justification:**

The authors addressed most of my concerns, but I was still not quite convinced about the choice of metrics.

Overall, I am in support of this paper being accepted, but I also feel there is room for improvement. Therefore, I maintain my original rating of Borderline Accept.

**Limitations:**

yes

**Paper Formatting Concerns:**

Minor concern, but whitespace seemed too low under some tables.

**Quality:**

3

**Strengths And Weaknesses:**

**Strengths**

1. The paper is well written and clear, with sufficient detail for the dataset construction. I liked the overall problem itself, and believe it can have a good impact on audiovisual research.
2. The experiments are well designed and thorough.
3. The dataset itself is a solid contribution and can also be useful for other audiovisual tasks beyond source localization.

**Weaknesses**

[Metric]

4. I am skeptical about the metric. I may be underestimating performance in many cases. If a human click is just a few pixels outside the segmentation map, I would argue this is still a correct (almost) prediction, but the metric would mark it as 0. Similarly, if the peak of predicted probabilities is even slightly off, it is marked as completely wrong.
5. The metric may also be overestimating performance, for example, in the ConflictVCue case of Fig. 8 where a prediction over the entire horse would be marked correct, even though its truly the horse’s head that should be marked correct.
6. The trend observed in Fig. 4 across different sizes may actually be because of the inherent bias in the metric itself. Bigger objects have more pixels that can be predicted to score correctly. The increase in the performance of the random predictor seems to be very similar to that of the models and human performance.

[Model]

7. Based on the heatmaps in Fig. 8, I think that IS3 results may not be a good datapoint to compare against. I would recommend the authors to revisit claims based on IS3 and maybe constrain their analysis to IS3-FT.
8. I am not familiar with the fixation cross. Can the authors include a brief description for why its needed.


[Results]

9. In Fig. 6, I could not see the dashed legend of vertical bias. May be a rendering issue at my end. Are the left ones for vertical and right ones for horizontal?
10. Was IS3 originally trained with stereo audio? Based on Table 1 (b) it seems like the gains from switching to stereo is somewhat consistent for IS3 and IS3+FT. If IS3 is not trained on stereo, I would find this trend curious and wonder if the performance difference between IS3 and IS3+FT is primarily due to smaller domain gap and not better utilization of stereo audio.

---

> ### Author Rebuttal · Authors · 2025-07-31
>
> Thank you for the detailed review and thoughtful feedback on our work. Below, we address the concerns point by point and respond to the reviewer’s questions individually.
>
> **f4Cu.1 - Questions: I am skeptical about the metric. I may be underestimating performance in many cases. If a human click is just a few pixels outside the segmentation map, I would argue this is still a correct (almost) prediction, but the metric would mark it as 0. Similarly, if the peak of predicted probabilities is even slightly off, it is marked as completely wrong.**
>
> We apply the same evaluation criterion to both human participants and AI models to ensure fair comparisons: a prediction is considered correct if the peak activation (for models) or the human click falls within the segmentation mask of the sounding object. In other words, all evaluations are conducted on equal footing.
> In response to the reviewer’s suggestion, we conducted an additional analysis by varying the pixel distance thresholds used to determine correctness. Specifically, a prediction is considered correct if it falls within x pixels of the ground-truth segmentation mask. We report the resulting A-Acc (accuracy with spatial tolerance) as a function of pixel thresholds in Table R1 below.
> From these results, we observe that while larger thresholds naturally lead to higher A-Acc values, the relative performance trend between humans and models remains consistent. This further supports the validity of our evaluation methodology.
> We will include these results and the corresponding discussion in the final version.
> Table R1. A-ACC as a function of thresholds under congruent conditions for object size2
> | Threshold      | Random | Human | IS3-FT-Stereo |
> |---------------:|:------:|:-----:|:-------------:|
> | 0 (default)    | 9.5    | 56.6  | 29.1          |
> | 10             | 9.6    | 57.4  | 29.9          |
> | 25             | 9.9    | 58.3  | 31.2          |
>
> **f4Cu.2 - Questions: The metric may also be overestimating performance, for example, in the ConflictVCue case of Fig. 8 where a prediction over the entire horse would be marked correct, even though its truly the horse’s head that should be marked correct.**
>
> A frog may emit vocalizations from its belly, a human from the throat, and a car from either the horn or the engine—highlighting that the exact sound-emitting region can vary across object categories. The reviewer raises an insightful point regarding fine-grained sound localization, which we agree is an important direction. We will include this consideration in the future work section.
> For the current study, we use the ground-truth segmentation mask of the entire sounding object, rather than attempting to isolate fine-grained vocalization regions, which are often unavailable in existing datasets.
> As mentioned in our earlier response to **f4Cu.1 - Questions**, we apply the same evaluation criterion to both human participants and AI models to ensure fair comparisons. While varying the evaluation criteria (e.g., localizing specific sound-emitting parts) could affect the absolute A-Acc values, we expect that the relative performance trends would remain consistent.
>
> **f4Cu.3 - Questions: The trend observed in Fig. 4 across different sizes may actually be because of the inherent bias in the metric itself. Bigger objects have more pixels that can be predicted to score correctly. The increase in the performance of the random predictor seems to be very similar to that of the models and human performance.**
> The reviewer correctly points out that raw A-Acc may be confounded by the area of the ground-truth mask—since, intuitively, smaller objects are more difficult to localize while larger ones are easier. To address this potential bias, we introduce a chance-corrected gain metric, which quantifies the improvement of a human or model over a random guess, normalized by the baseline accuracy of the random guess:
>
> $$
> \mathrm{Gain}(X) = \frac{\mathrm{Acc} _X - \mathrm{Acc} _{\text{rand}}}{\mathrm{Acc} _{\text{rand}}} \times 100 \%
> $$
>
> $\displaystyle \text{Acc}_{\text{rand}}$ represents the accuracy achieved when predictions are sampled uniformly at random across the scene.
>
> This normalization removes the influence of mask size and isolates the true localization capability of humans and models. As shown in **Table R2** below, even after correcting for chance, humans consistently outperform models—especially for small object sizes. For example, in the smallest size category, humans achieve a gain of 17.9%, compared to only 1.7% for models.
> These results suggest that object size modulates sound source localization performance in a way that cannot be fully explained by ground-truth mask area alone, highlighting deeper perceptual and representational differences between human and model behavior.
> Table R2. Chance-corrected gain for humans and models across object sizes
> | Normalized Percentage  | Size 1 | Size 2 | Size 3 |
> |------------------------|:------:|:------:|:------:|
> | (model − random) / random | 1.7   | 0.4   | 0.4   |
> | (human − random) / random | 19.6  | 5.0   | 2.2   |
> | (human − model) / random  | 17.9  | 4.5   | 1.7   |
>
> **f4Cu.4 - Questions: Based on the heatmaps in Fig. 8, I think that IS3 results may not be a good datapoint to compare against. I would recommend the authors to revisit claims based on IS3 and maybe constrain their analysis to IS3-FT.**
>
> We chose IS3 because it is a widely used baseline in the literature and achieves the best performance among all baselines under most of the conditions. Therefore, we use IS3 as the representative baseline for the main paper’s analyses. Additionally, we compare other baselines against humans and IS3-FT-Stereo, providing these results and analyses in the appendix due to space constraints.
> As suggested by the reviewer, we will revise Figure 8 to include more visualization results from other competitive baselines such as CAVP, and we will reduce the amount of discussion focused solely on IS3 in the final version.
>
> **f4Cu.5 - Questions: I am not familiar with the fixation cross. Can the authors include a brief description for why its needed.**
>
> The pre-trial fixation dot is a crucial experimental design element commonly employed in classical human psychophysical studies within neuroscience, cognitive science, and psychology. At the start of each trial, it recenters participants’ attention by requiring them to fixate at the center of the screen. This ensures that attention originates from a common spatial and attentional baseline for every trial, thereby eliminating potential carry-over effects from the preceding trial. As a result, any observed differences in response latency or eye movement trajectories can be confidently attributed to the experimental manipulation.
> In our human psychophysics experiments, we use eye-tracking devices to monitor eye movements for attention control—an important detail not mentioned in the current version due to space constraints. We will include this information about our eye-tracking data quality control in the final version.
>
> **f4Cu.6 - Questions: In Fig. 6, I could not see the dashed legend of vertical bias. May be a rendering issue at my end. Are the left ones for vertical and right ones for horizontal?**
>
> This issue is likely due to rendering differences caused by local machine configurations. In each group of the bar plots, the left bar, which features slanted line textures on its face, represents vertical localization performance, while the right bar, without textures, corresponds to horizontal localization precision. To avoid similar rendering problems on other systems, we will update the figure’s presentation style in the final version.
>
>
> **f4Cu.7 - Questions: Was IS3 originally trained with stereo audio? Based on Table 1 (b) it seems like the gains from switching to stereo is somewhat consistent for IS3 and IS3+FT. If IS3 is not trained on stereo, I would find this trend curious and wonder if the performance difference between IS3 and IS3+FT is primarily due to smaller domain gap and not better utilization of stereo audio.**
>
> We introduce three IS3 variants in the paper:
> First, IS3: The original model trained on the Flickr-SoundNet [53] and VGG-Sound [46] datasets.
> Second, IS3-FT-mono: The original model fine-tuned on our AudioCOCO dataset, using monaural audio as input.
> Third, IS3-FT-Stereo: The original model fine-tuned on our AudioCOCO dataset, using stereo audio channels as input.
> The original IS3 model exhibits a bias toward large, centrally located objects. As discussed in the Introduction and illustrated in Supplementary Figures S1–S3, our proposed filtering method applied to the AudioCOCO dataset mitigates this bias, resulting in a higher-quality dataset drawn from the same source domain.
> The observed performance improvements in IS3-FT-Stereo can thus be attributed to both the enhanced data quality and the inclusion of stereo audio. The performance gap between IS3-FT-Mono and the original IS3 model is 3.3% for Size1 (small objects), 7.7% for Size2, and 9.3% for Size3, clearly demonstrating the effect of improved data quality.
> Furthermore, the comparison between IS3-FT-Mono and IS3-FT-Stereo shows gains of 4.4% for Size1, 4.4% for Size2, and 1.4% for Size3, highlighting the contribution of spatial cues derived from stereo audio, rather than domain gap effects.
> Together, these results validate the effectiveness of our dataset refinement and the utility of stereo input for improved spatial localization. We will clarify these IS3 model variants in the final version.

---

> > ### Comment · Reviewer_f4Cu · 2025-08-06
> > **Response to rebuttal**
> >
> > I appreciate the authors’ thoughtful and detailed response.
> >
> > **Evaluation Metric**
> >
> > I am still skeptical of the metric even with the new soft thresholding experiments presented by the authors. Applying the same criterion to AI models and humans does not necessarily equate to a fair comparison because of the inherent difference between them. When a human clicks a point, they don't necessarily look at all points in the image and pick the "best".
> >
> > However, I do not have a concrete alternative and understand that the authors’ choice of metrics may be the most appropriate and sufficient for the claims in this work.
> >
> > **Performance vs size of Object**
> >
> > Thank you for tabulating the normalized percentages. I agree with the authors' interpretation.
> >
> > ---
> >
> > Thank you for the other clarifications, most of my comments have been addressed.

---

> > > ### Author Response · Authors · 2025-08-08
> > >
> > > Thank you for the feedback!
> > >
> > > We agree that a truly fair comparison between AI models and humans is inherently difficult. Humans benefit from years of real-world experience and interact with the world through foveated vision, while AI models are typically trained on internet-scale datasets and process all pixels uniformly. Nevertheless, benchmarking AI models against human performance is essential for advancing both AI and neuroscience.
> > >
> > > To facilitate meaningful comparisons, we adopted a common and quantifiable metric: whether the click falls within the segmentation mask—a criterion applicable to both humans and AI models (see response **f4Cu.1 - Questions**).  Additionally, in response to the reviewer’s suggestion, we evaluated all AI models using the center-weighted Intersection over Union (cIoU) metric, which is widely used in the object detection literature. Under this evaluation, our IS3-FT-Stereo model still outperforms other models (see **BL6m.2 - Weakness** for details).
> > >
> > > We will incorporate these clarifications in the Discussion section and include the additional cIoU-based experimental results in the revised manuscript.

---

### Decision · Program_Chairs · 2025-09-17

**Decision:**

Accept (spotlight)

**Comment:**

(a, b) There are a number of strengths and impacts of this paper. The problem of understanding modeling and inductive/task bias between audio and visuals is not well understood in the multimodal learning literature. AV literature is very broad but some of the key points that authors raise -- human vs. non-human evaluation mismatches that drive model bias; lack of a controlled task and relevant data where various combinations of AV scenarios can be systematically tested for realism as well as representation-driven performance. It is interesting to see that localization task fit this quite well (this in itself is a surprising finding). The evaluations are done well; although there have been some questions about the metric, the rationale for using it makes sense.

(c) More larger scale evaluations of the proposed algorithm/model especially on the traditional AV benchmarks could have boosted the paper. Bulk of the evaluations are centered on the dataset that the authors created. Bulk of the findings from mono vs. binaural/stereo are well known in psychoscoustics (e.g., ITD/ILD related) and so these are not surprising findings as such.

(d) This is a well conducted study to understanding the representational power of audio vs. visual in joint audio-visual tasks. Separating the modeling bias from dataset bias by creating a novel task and an associated benchmark that allows interpreting the failure cases (eg., object sizes impact localization accuracy) and identifying psychocoustic outcomes (most of which are known, but atleast its retroactive evidence) is useful. Using human vs. non-human evals as a direct measure for performance is interesting.

(e) Reviewers mostly agreed on the strengths and impact; and the rigor of execution in the paper. Bulk of their concerns were regarding miss interpretation / lack of clarity on metrics; potential limitations of dataset generality and missing details re: related work etc. All these are answered.